# Itinerant ferromagnetism in dilute SU(N) Fermi gases

Jordi Pera, Joaquim Casulleras and Jordi Boronat

Departament de Física, Universitat Politècnica de Catalunya,
Campus Nord B4-B5, E-08034 Barcelona, Spain

## Abstract

We present exact analytic results for the energy of a SU(N) repulsive Fermi gas as a function of the spin-channel occupation at second order in the gas parameter. This is an extension of previous results that now incorporates the degree of polarization of the system. Therefore, the magnetic properties of the gas can be obtained, free from numerical uncertainties. For spin 1/2 we find that second-order corrections change the itinerant ferromagnetic transition from continuous to first-order. Instead, for spin larger than 1/2 the phase transition is always of first-order type. The transition critical density reduces when the spin increases, making the phase transition more accessible to experiments with ultracold dilute Fermi gases. Estimations for Fermi gases of Yb and Sr with spin 5/2 and 9/2, respectively, are reported.



# 1 Introduction

At low density an electron gas is paramagnetic rather than ferromagnetic because its energy is essentially kinetic. At a same density, the Fermi energy of a polarized (ferromagnetic) phase is larger than the one of the unpolarized (paramagnetic) one [1]. However, increasing the density produces an increase of the potential energy due to the interaction of electrons with different spin. At some point, this increment of potential energy can exceed the kinetic energy gap and the gas will become ferromagnetic. This is the well-known Stoner scenario of itinerant ferromagnetism [2] where a quantum magnetic transition is predicted in the absence of any crystal pattern and any magnetic field.

However, the observation of itinerant ferromagnetism in different materials has been extremely elusive because an increase in the density can produce non-Fermi liquids [3] or crystallization [4] before the expected ferromagnetic transition. The high tunability of trapped cold Fermi gases has offered a new platform to study itinerant ferromagnetism. However, even in this case, the observation of this transition has turned elusive because the repulsive branch is metastable with respect to the formation of spin up-spin down dimers. A first pioneering observation of the ferromagnetic transition [5] was later revised and concluded that pair formation precludes the achievement of the relatively large gas parameters required for observing this transition [6]. More recently, it has been claimed that the ferromagnetic state is effectively obtained after the observation of spin-domain inmiscibility in Fermi $^6$Li around a gas parameter $x = k_F a_0 \simeq 1$, with $k_F$ the Fermi momentum and $a_0$ the s-wave scattering length [7]. The theoretical study of repulsive Fermi gases, to determine the gas parameter at which the ferromagnetic transition appears, has been intense [8–15]. Itinerant ferromagnetism is predicted to happen around $x \simeq 1$, with the most reliable results derived from quantum Monte Carlo (QMC) simulations [9]. At this value of the gas parameter, the Stoner model is not quantitatively accurate and the use of microscopic approaches as QMC is the best tool in spite of the sign problem.

The Stoner model is derived using first-order perturbation theory on the gas parameter, that is it is the Hartree-Fock model for the repulsive Fermi gas [2]. For particles with spin 1/2, this model predicts a continuous transition from a paramagnetic phase to a ferromagnetic one at $x = \pi/2$. The transition point obtained with QMC is significantly smaller [9], pointing to the relevance of interactions which Stoner model incorporates only at first order. The first correction to the Hartree-Fock energy was obtained by Huang, Lee, and Yang using pseudopotentials for the hard-sphere Fermi gas [16, 17]. The same result was also derived by Galitskii [18], using Green functions, and Abrikosov and Khalatnikov by means of Landau theory of Fermi liquids [19]. This analytical result for the second-order perturbation theory gives the total energy of the Fermi gas as a single function of the gas parameter. However, one cannot use this equation when the number of particles in each spin is different or, in other words, the polarization does not enter in it as a variable.

In recent years, the experimental production of SU(N) fermions has renewed the theoretical interest in their study. Cazalilla *et al.* [20] discussed that Fermi gases made of alkaline atoms with two electrons in the external shell, such as $^{173}$Yb, present an SU(N) emergent symmetry. They also claimed that for $s > 1/2$ the ferromagnetic transition had to be of first order because the mathematical structure of SU(N>2) is significantly different than the one of SU(2). In 2014, Cazalilla and Rey [21] reviewed the progress made with ultra-cold alkaline-earth Fermi gases. Ref. [22] was one of the first observations of SU(N) symmetric interactions in alkaline-earth-metal atoms. Interaction effects in SU(N) Fermi gases as a function of N were studied in Ref. [23]: for weak interactions it was predicted that inter-particle collisions are enhanced by N. Collective excitations in SU(N) Fermi gases with tunable spin proved to be instrumental to investigate collective properties of large spin systems [24]. On the other hand,

Ref. [25] studied the prethermalization of these systems starting from different initial conditions finding that, under some conditions, the imbalanced initial state could be stabilized for a certain time. Recently, Ref. [26] performed a thorough study of the thermodynamics of deeply degenerate SU(N) Fermi gases using $^{87}$Sr for which N can be tuned up to 10. For temperatures above the super-exchange energy, the behavior of the thermodynamic quantities was found to be universal with respect to N [27].

In the present work, we solve analytically the second-order perturbation energy including the polarization or, more precisely, the dependence on the relative concentration of particles with different $z$-spin component. Our results are derived for a generic spin and thus, we can apply it to SU(N) fermions. This generalization allows, for instance, the study of dilute Fermi gases of Ytterbium [28], with spin 5/2, and Strontium [29], with spin 9/2, that have been already produced in experiments. With the analytic result for the energy, as a function of the gas parameter and occupation of spin channels, we perform a Landau analysis of the ferromagnetic transition for any spin. We find that, for spin 1/2, the phase transition turns to be first-order with respect to the polarization instead of the continuous character of it derived in the Stoner model. Interestingly, the critical density for itinerant ferromagnetism is observed to decrease monotonously when the spin increases, opening new possibilities for experimental realizations in cold Fermi gases.

## 2 Methodology

We study a repulsive Fermi gas at zero temperature with spin $S$ and spin degeneracy $\nu = 2S+1$. In the dilute gas regime, only particles with different $z$-spin component interact via a central potential $V(r)$ ($s$-wave scattering). The number of particles in each spin channel is $N_\lambda = C_\lambda N/\nu$, with $N$ the total number of particles and $C_\lambda$ being the fraction of $\lambda$ particles (normalized to be one if the system is unpolarized, $N_\lambda = N/\nu$, $\forall \lambda$). The Fermi momentum of each species is $k_{F,\lambda} = k_F C_\lambda^{1/3}$, with $k_F = (6\pi^2 n/\nu)^{1/3}$. The kinetic energy is readily obtained,

$$\frac{T}{N} = \frac{3}{5}\epsilon_F \frac{1}{\nu} \sum_\lambda C_\lambda^{5/3}, \tag{1}$$

with $\epsilon_F = \hbar^2 k_F^2/(2m)$ the Fermi energy. The lowest-order contributions to the potential energy are given by [30]

$$V = \frac{\hbar^2 \Omega}{2m} \sum_{\lambda_1,\lambda_2} \int \frac{d\mathbf{l}}{(2\pi)^3} n_l \int \frac{d\mathbf{k}}{(2\pi)^3} n_k \{K(\mathbf{k},\mathbf{l};\mathbf{k},\mathbf{l}) - \delta_{\lambda_1,\lambda_2} K(\mathbf{k},\mathbf{l};\mathbf{l},\mathbf{k})\}, \tag{2}$$

with $\Omega$ the volume, $n_l$ and $n_k$ the momentum distributions of the free Fermi gas. Up to second order in the $s$-wave scattering length $a_0$, the scattering $K$ matrix is given by [30,31]

$$K(r,R) = 4\pi a_0 + (4\pi a_0)^2 I(r,R) + O(a_0^3), \tag{3}$$

with $\mathbf{r} = (\mathbf{k}-\mathbf{l})/2$ and $\mathbf{R} = \mathbf{k}+\mathbf{l}$, the relative and total momentum in the center of mass frame, respectively. The function $I(r,R)$ is defined as

$$I(r,R) = \frac{1}{(2\pi)^3} \int 2\,d\mathbf{q}d\mathbf{q}' \frac{1-(1-n_q)(1-n_{q'})}{q^2+q'^2-k^2-l^2} \delta(\mathbf{q}+\mathbf{q}'-\mathbf{k}-\mathbf{l}). \tag{4}$$

Considering only the first term in the expansion of the $K$-matrix (3), one gets for the potential energy (2) the well-know Hartree-Fock energy [2],

$$\left(\frac{V}{N}\right)_1 = \frac{2\epsilon_F}{3\pi} \left[\frac{1}{\nu} \sum_{\lambda_1,\lambda_2} C_{\lambda_1} C_{\lambda_2}(1-\delta_{\lambda_1,\lambda_2})\right] x, \tag{5}$$

with $x \equiv k_F a_0$ the gas parameter of the Fermi gas.

The second order term in the gas parameter $x$ is due to the second term of the $K$-matrix. This second order term reads

$$\left(\frac{V}{N}\right)_2 = \frac{\epsilon_F}{k_F^7}\left[\frac{1}{\nu}\sum_{\lambda_1,\lambda_2} I_2(k_{F,\lambda_1}, k_{F,\lambda_2})(1-\delta_{\lambda_1,\lambda_2})\right]x^2, \tag{6}$$

with

$$I_2(C_{\lambda_1}, C_{\lambda_2}) = \frac{3}{16\pi^5}\int d\mathbf{l}\, n_l \int d\mathbf{k}\, n_k \int 2\, d\mathbf{q}\, d\mathbf{q}'\frac{1-(1-n_q)(1-n_{q'})}{q^2+q'^2-k^2-l^2}\delta(\mathbf{q}+\mathbf{q}'-\mathbf{k}-\mathbf{l}). \tag{7}$$

The calculation of $I_2(C_{\lambda_1}, C_{\lambda_2})$ (7) is rather involved. For the unpolarized phase, i.e., when all the spin states are equally populated, the integral in Eq. (7) was made in the fifties of the past century [16–19]. On the other hand, when the gas has a finite polarization that integral becomes more cumbersome. In previous work, it was solved partially but with a final numerical integration [32]. We have been able to integrate Eq. (7) and found an analytical expression for it (See App. C). Our result is the following,

$$I_2(C_{\lambda_1}, C_{\lambda_2}) = \frac{4k_F^7}{35\pi^2}C_{\lambda_1}C_{\lambda_2}\frac{C_{\lambda_1}^{1/3}+C_{\lambda_2}^{1/3}}{2}F(y), \tag{8}$$

with

$$F(y) = \frac{1}{4}\left(15y^2 - 19y + 52 - 19y^{-1} + 15y^{-2}\right) + \frac{7}{8}y^{-2}(y-1)^4(y+3+y^{-1})\ln\left|\frac{1-y}{1+y}\right|$$
$$-\frac{2y^4}{1+y}\ln\left|1+\frac{1}{y}\right| - \frac{2y^{-4}}{1+y^{-1}}\ln\left|1+y\right|, \tag{9}$$

and $y \equiv (C_{\lambda_1}/C_{\lambda_2})^{1/3}$. Our result reproduces the formula derived by Kanno [33] for the specific case of $s = 1/2$ and hard spheres.

Assembling it all together, we can write the energy per particle up to second order in $x$, and for any occupation of the $\nu$ available spin states, as

$$\frac{E}{N} = \frac{3\epsilon_F}{5\nu}\left\{\sum_\lambda C_\lambda^{5/3} + \frac{10}{9\pi}\left[\sum_{\lambda_1,\lambda_2}C_{\lambda_1}C_{\lambda_2}(1-\delta_{\lambda_1,\lambda_2})\right]x + \frac{5}{3k_F^7}\left[\sum_{\lambda_1,\lambda_2}I_2(C_{\lambda_1},C_{\lambda_2})(1-\delta_{\lambda_1,\lambda_2})\right]x^2\right\}, \tag{10}$$

with $I_2(C_{\lambda_1}, C_{\lambda_2})$ given by Eqs. (8,9). Eq. (10) is a perturbative expansion in $x$ and works fine for low values of $x$, $x < 1$. If the gas is unpolarized, i.e., $C_\lambda = 1, \forall\lambda$, the energy per particle reduces to the known expression [30],

$$\frac{E}{N} = \frac{3\epsilon_F}{5}\left\{1+(\nu-1)\left[\frac{10}{9\pi}x + \frac{20}{105\pi^2}(11-2\ln2)x^2\right]\right\}. \tag{11}$$

The energy of the interacting Fermi gas (10) is written in terms of the occupation of the different spin channels $C_\lambda$. This set of values results in a polarization $P$ of the system, in such a way that when all the spin states are equally populated the gas is unpolarized $P = 0$ and, if only one of them is populated, $|P| = 1$. Keeping the total number of particles $N$ as constant, and for $s = 1/2$, there is only one ratio of spin occupations for a given value of $P$. In contrast, for $s > 1/2$ there are more combinations. It can be shown that the solution which optimizes the energy is the one in which the increase of particles in one spin state comes for an equal

decrease of the rest with constant $N$ (See App. A). Under these conditions, the concentrations $C_\lambda$ for a given polarization $P$ are

$$C_+ = 1 + |P|(\nu - 1), \tag{12}$$

$$C_{\lambda \neq +} = 1 - |P|, \tag{13}$$

with subindex $+$ standing for the state with the larger population. As we have discussed, for $s > 1/2$ there are more possible configurations. For example, Ref. [25] works with a system of $s = 3/2$ with population imbalance: the number of atoms with $s = 3/2$, $N_{\pm 3/2}$, is different than the one with $s = 1/2$, $N_{\pm 1/2}$. As we need to make a choice, we choose the one that minimizes the energy.

It is interesting to check if Eq. (10) converges when the degeneracy increases. If we set the limit $\nu \to \infty$ in Eq. (10), we get

$$\frac{E}{N} = \frac{2\pi\hbar^2}{ma_0^2}\left[\frac{3}{20\pi}(6\pi^2 n)^{2/3}a_0^2 P^{5/3} + na_0^3(1 - P^2) + \frac{1}{2\pi^3}(6\pi^2 n)^{4/3}a_0^4(1 - P)P^{4/3}\right]. \tag{14}$$

As one can see, Eq. (14) is convergent, as it is finite for any value of $P$. Particularizing to the unpolarized case $P = 0$, one obtains the Hartree-Fock energy for bosons. And, if we set $P = 1$, we obtain the energy of a non-interacting Fermi gas.

## 3 Landau Theory

Upon an increase of the gas parameter, the interacting Fermi gas will eventually become ferromagnetic. To localize the transition point and the order of the phase transition we rely on Landau theory. We first explore what we obtain for the Stoner model, that is, first order approximation in $x$. Then, we consider the second-order expansion, Eq. (10). In this way, we can compare the differences that arise due to increasing the order of the perturbative expansion. At the Hartree-Fock level, first order in $x$, the Landau expansion of the energy is given by

$$f - f_0(x) = -\frac{1}{2}A(x - x_0)P^2 - \frac{1}{3}B|P|^3 + \frac{1}{4}CP^4, \tag{15}$$

with $f = 5E/(3N\epsilon_F)$, $x_0 = \pi/2$, and the rest of constants are given in App. B. In order to determine the transition point, one imposes two conditions: $i$) the energy has to be a minimum, that is, its first derivative must be zero, and $ii$) the energy must be smaller than the zero-polarization energy to have a global minimum. With these two criteria, one finds $x^* = x_0 - 2B^2/(9AC)$ and $|P^*| = 2B/(3C)$. The constant $B$ is proportional to $(\nu - 2)$, therefore the Stoner model predicts a first-order phase transition for spin higher than $1/2$, while a continuous one for $s = 1/2$.

Equipped with the second-order expression of the energy as a function of the spin-state occupations (10), we can make a Landau expansion of the energy. We get

$$f - f_0(x) = -\frac{1}{2}A(\overline{x}(x) - x_0)P^2 - \frac{B(x)}{3}|P|^3 + \frac{C(x)}{4}P^4 + \frac{L(x)}{4}P^4 \ln|P|. \tag{16}$$

The coefficients in Eq. (16) depend now on the gas parameter $x$ and have a more complex expression (See App. B). For the sake of simplicity, from now on, we will not specify that the coefficients of Eq. (16) depend on the gas parameter $x$. Applying the same criteria for finding the transition point as before, we derive the equations for the jump of polarization and transition density values,

$$\ln|P^*| = -\left(\frac{1}{2} + \frac{C}{L}\right) + \frac{2B}{3L|P^*|}, \tag{17}$$

$$\overline{x}^* = x_0 - \frac{B}{3A}|P^*| - \frac{L}{4A}|P^*|^2. \tag{18}$$

We rearrange Eq. (17) to explore the range where a solution exists,

$$\frac{2B}{3L|P^*|} - \ln|P^*| = \frac{1}{2} + \frac{C}{L}. \tag{19}$$

The function $2B/(3L|P^*|) - \ln|P^*|$ is continuously decreasing between $|P^*| = 0$ and $|P^*| = 1$ as long as $2B/(3L)$ is positive. As this is the case, in this range, the minimum value will be at $|P^*| = 1$. Hence, there will exist a solution when the coefficients satisfy the condition

$$\frac{1}{2} + \frac{C}{L} - \frac{2B}{3L} \geq 0. \tag{20}$$

This condition is fulfilled for any spin value and for densities lower than the transition one. For spin $1/2$, the above equations suffer an important simplification as the coefficient $B$ is zero,

$$\begin{aligned}
|P^*| &= \exp\left(-\frac{1}{2} - \frac{C}{L}\right), \\
\overline{x}^* &= x_0 - \frac{L}{4A}\exp\left(-1 - \frac{2C}{L}\right).
\end{aligned} \tag{21}$$

It is worth noticing the relevance of the coefficient $L$ in Eq. (21), coming from the term $P^4\ln|P|$, for distinguishing the order of the phase transition: $L$ needs to be different from zero to get a first-order phase transition [34]. Introducing the values of the constants in Eq. (21), one gets the transition density at $x = \pi/3(1 + 0.007) \approx 1.0545$, and a jump in the polarization equal to 0.545. Remarkably, the second-order energy changes the character of the ferromagnetic phase transition becoming now a first-order one, as it happens for spin $s > 1/2$.

## 4 Results

We have applied the above formalism to study fundamental properties of the Fermi gas, such as the energy or the magnetic susceptibility.

In Fig.1, we show the dependence of the polarization $P$ with the gas parameter $x$ for a $s = 1/2$ Fermi gas. One can observe as the introduction of second-order corrections to the potential energy modify the character of the phase transition with respect to the Stoner model. Moreover, the gas parameter at which the transition occurs is significantly reduced, from $\pi/2$ to $\sim \pi/3$, approaching the value $x \simeq 1$ where the transition has been indirectly observed [7].

In Fig. 2, we plot the energy of the Fermi gas using the the Stoner one and the second-order approximation. At very low densities, both models predict the same energy, which follow the non-polarized behavior. After a certain value of the gas parameter, the lines become flat, which is the behavior of the fully-polarized gas. We can see that, when we use the second-order approximation, the transition happens at a smaller gas parameter.

The first-order phase transition for $s = 1/2$ is also observed in Fig. 3. There, we plot the energy as a function of $P$ and for gas parameter values close to the phase transition. One can see that the minimum of the energy jumps from $P = 0$ to an intermediate value $P = 0.545$ that then progressively moves to the fully polarized phase (Fig. 1).

Since we are interested in the magnetism of these gases, we proceed to analyze the magnetic susceptibility, which is inversely proportional to the second derivative of the energy with respect to the polarization,

$$\frac{1}{\chi} = \frac{1}{n}\left(\frac{\partial^2(E/N)}{\partial P^2}\right)_x. \tag{22}$$



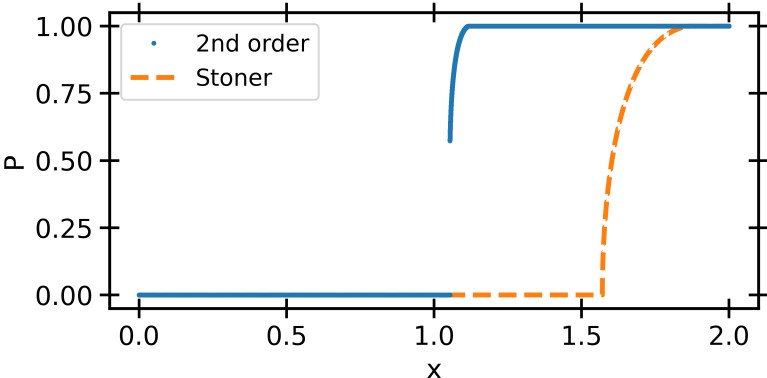

Figure 1: Itinerant ferromagnetic transition for $s = 1/2$. The dashed orange and solid blue lines stand for the Stoner and second-order approximations, respectively.

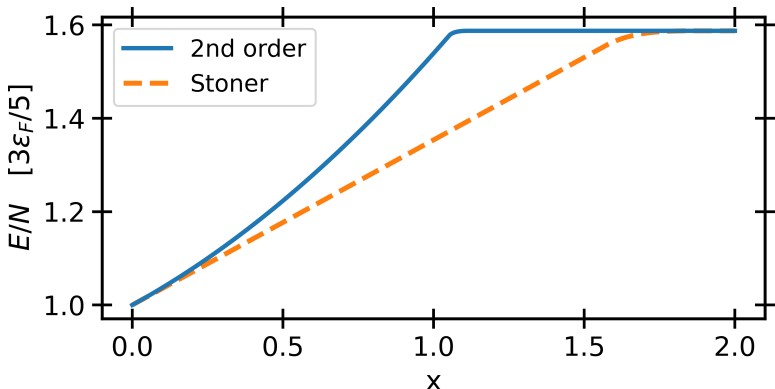

Figure 2: Energy per particle as a function of the gas parameter for $s = 1/2$. The dashed orange and solid blue lines stand for the Stoner and second-order approximations, respectively.

If we split the total energy between the kinetic and the potential energy, we can rewrite $\chi$ as

$$\chi = \frac{3}{2} \frac{n}{\epsilon_F} \left[ \frac{1}{\nu} \sum_\lambda C_\lambda^{-1/3} (C_\lambda')^2 + \frac{3}{2} \frac{1}{\epsilon_F} (V/N)'' \right]^{-1}, \tag{23}$$

where the derivatives are with respect to the polarization. In Fig. 4, we show the magnetic susceptibility around the ferromagnetic transition point for $s = 1/2$. We see again that the transition occurs at a lower value of $x$ [9] with respect to the Stoner model. One can also notice that $\chi$ changes behavior, from diverging at the transition point (Stoner) to a large but finite peak at second order, reflecting the change in the type of phase transition from a continuous to a first-order one.

From Eq. (23), one can prove that at first order $\chi$ diverges. At this order, $\chi$ behaves around the transition as

$$\chi = \frac{3}{2} \frac{n}{\epsilon_F} \left[ -\frac{9}{10} A(x - x_0) \right]^{-1} = \frac{3}{2} \frac{n}{\epsilon_F} \frac{5C}{B^2}. \tag{24}$$

The coefficient $B$ in Eq. (24) is proportional to $(\nu - 2)$ and $C$ is finite for $\nu = 2$ (See App. B), hence, the magnetic susceptibility diverges around the transition for the Stoner model.

We can extend our analysis to larger spin Fermi gases. In this case, the ferromagnetic transition is first-order as in the Stoner model but the transition point is, in all cases, observed

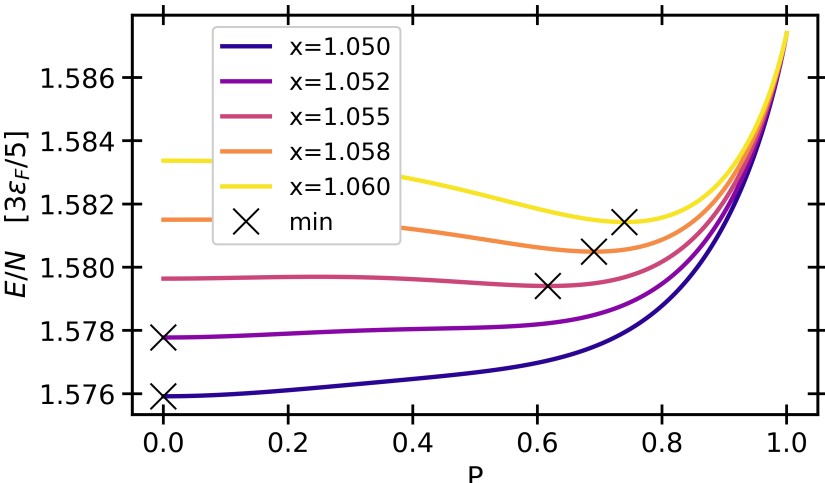

Figure 3: Energy per particle as a function of the polarization $P$ for $s = 1/2$. The lines correspond to different $x$ values close to the phase transition $x^\star = 1.054$. The crosses in each line indicates the polarization where the energy is minimum.

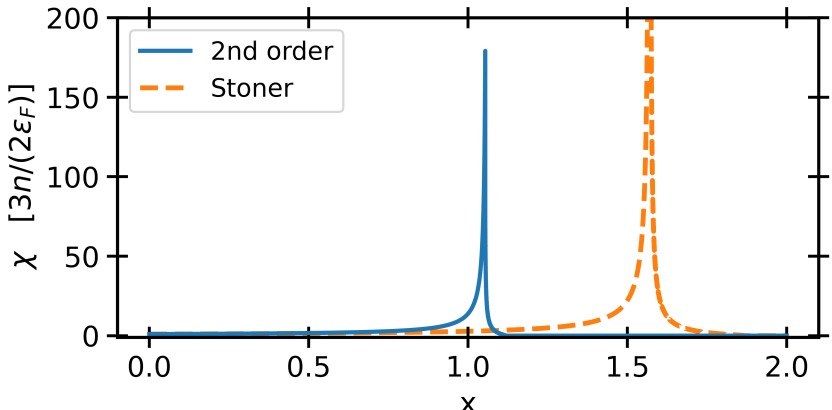

Figure 4: Magnetic susceptibility $\chi$ around the transition point for $s = 1/2$. The peak of $\chi$ appears at a lower $x$ value than in the Stoner model.

at smaller $x$ values. This fact can be seen in Fig. 5 where the energy of the spin 9/2 Fermi gas changes its behavior at a lower gas parameter than the energy of the spin 5/2 Fermi gas.

In Fig. 6, we show the energies of SU(N) Fermi gases up to second order as a function of the polarization for $x$ values close to the phase transition. The top panel is for $s = 5/2$ and the bottom one for $s = 9/2$ corresponding to Yb and Sr, respectively. In both cases, the location of the minimum of the energy as a function of the polarization jumps from $P = 0$ to 1 without intermediate values, in contrast to the case of $s = 1/2$. For the sake of completeness, in Fig. 7 we show the magnetic susceptibility for the two high degenerate Fermi gases considered: $s = 5/2$ and $s = 9/2$.

Thanks to having an analytical expression for the energy, we have access to other important properties as the Tan's contact. According to Tan relations, the microscopic behavior of the wave function of the $N$-body system at short distances ($r \ll n^{-1/3}$) is connected with the behavior of several macroscopic magnitudes. In particular, the tail of the momentum distribution for large $\mathbf{k}$ values as $k^{-4}$ and the dependence of the energy on the scattering length. Using the adiabatic sweep theorem [35], the Tan's contact can be obtained from the energy of

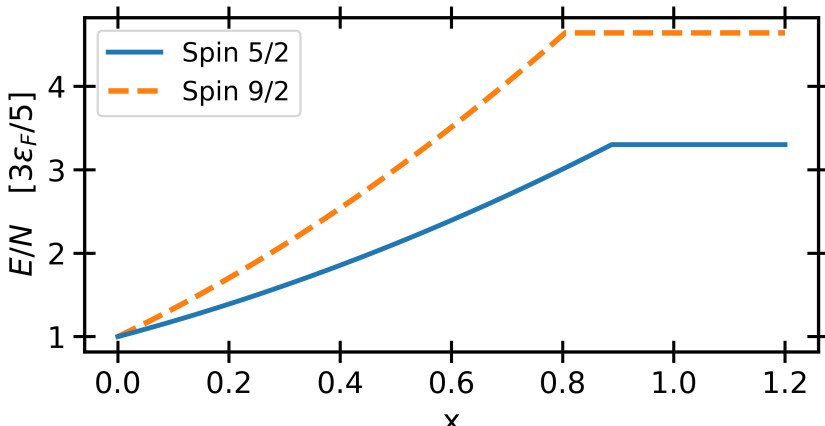

Figure 5: Energy per particle as a function of the gas parameter. The solid blue and dashed orange lines stand for a Fermi gas of $s = 5/2$ and $s = 9/2$, respectively. Both lines have been calculated within the second-order approximation.

the system by

$$C = \frac{8\pi m a_0^2}{\nu \hbar^2} \frac{N}{V} \frac{\partial (E/N)}{\partial a_0} . \tag{25}$$

In terms of the gas parameter,

$$C = \frac{4\pi n k_F}{\nu} \frac{x^2}{\epsilon_F} \frac{\partial (E/N)}{\partial x} . \tag{26}$$

We have calculated the Tan's contact using the energy of the Fermi gas at second order. In Fig. 8, we show $\nu C$ as a function of $x$ for spin $s = 1/2$, 5/2, and 9/2. In the paramagnetic phase, $\nu C$ increases monotonically until it reaches its maximum value at the transition point. After crossing the ferromagnetic transition, the Tan's contact becomes zero because the energy of the fully polarized phase does not depend on the scattering length $a_0$. As one can see in the figure, the $\nu C$ increases with the value of the spin for a given $x$ value, in agreement with the increase of interaction energy with spin degeneracy. It is interesting to notice that, for spin 1/2, $C$ does not drop abruptly to zero because there are stable polarizations between 0 and 1. In contrast, for $s > 1/2$ the drop is directly to zero because of the sudden change of the polarization from 0 to 1 at the phase transition.

In Fig. 9, we show the evolution of the Tan's contact as a function of the spin for the Stoner and second-order models. By increasing the spin, one can see a tendency to reach a plateau. This plateau, which has been interpreted as the Bose limit, can be understood by looking at the Tan's contact behavior before the transition. For the Stoner model, it is given by

$$C = 4\pi n k_F x^2 \frac{2}{3\pi} \left( 1 - \frac{1}{\nu} \right), \tag{27}$$

and in second-order,

$$C = 4\pi n k_F x^2 \left[ \frac{2}{3\pi} + \frac{8}{35\pi^2} (11 - 2\ln 2) x \right] \left( 1 - \frac{1}{\nu} \right). \tag{28}$$

The dependence with $\nu$ for both models is $1 - 1/\nu$, hence, if we set the limit $\nu \to \infty$, we obtain a plateau. In Fig. 9, we also plot experimental results from Ref. [36] that show a similar behavior to our theoretical expressions (Eqs. (27) and (28)). Moreover, they lie between both

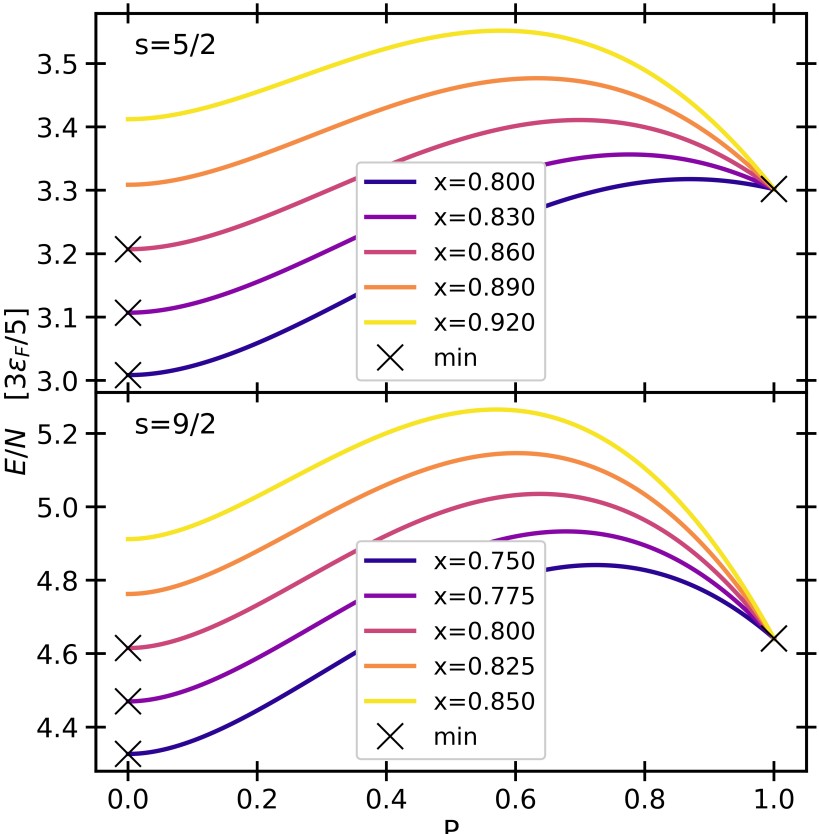

Figure 6: Energy of the $s = 5/2$ (top) and $s = 9/2$ (bottom) Fermi gases as a function of $P$ for $x$ values close to the transition. The crosses indicate the values of $P$ where the energy is minimum.

models, but we cannot state which one fits better due to the experimental uncertainty. As the experimental points have a different normalization than our definition, we have to scale them by a factor of $(2\pi)^2$.

The critical value of the gas parameter depends on the spin of the particles. In Fig. 10, we report the results obtained for both models (Stoner and second order). As one can see, the itinerant ferromagnetic transition happens at $x$ values that decrease monotonically with the spin degeneracy. Moreover, the second order values of $x$ are lower than the ones predicted by the Stoner model. As for $s > 1/2$ the polarization goes from 0 to 1 without intermediate values ($P^* = 1$), we can find more easily the laws that the critical values $x^*$ follow. For the Stoner model, the law is given by

$$x^* = \frac{9\pi}{10(\nu-1)}\left(\nu^{2/3} - 1\right),\tag{29}$$

and in second-order,

$$x^* = \pi \frac{-35 + 35\sqrt{1 + \frac{108(11 - 2\ln 2)}{175(\nu-1)}\left(\nu^{2/3} - 1\right)}}{12(11 - 2\ln 2)}.\tag{30}$$

Notice that for $s = 1/2$ Eqs. (29) and (30) do not hold. In this case, the value of $x^*$ is slightly smaller than the value predicted by the laws (29,30) because the transition is from $P = 0$ to

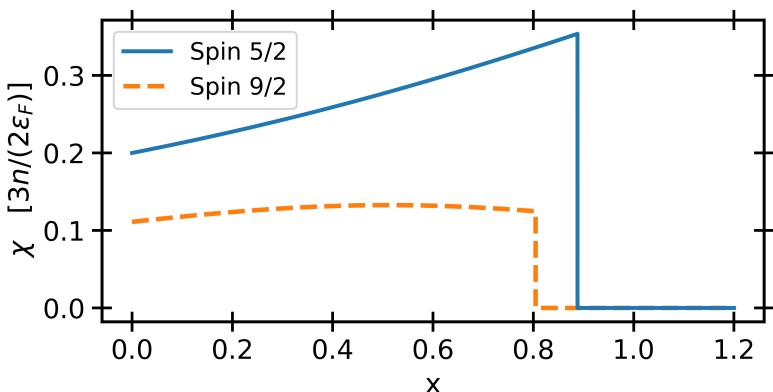

Figure 7: Magnetic susceptibility per particle as a function of the gas parameter. The slid blue and dashed orange lines stand for a Fermi gas of $s = 5/2$ and $s = 9/2$, respectively. Both lines have been computed with the second-order approximation.

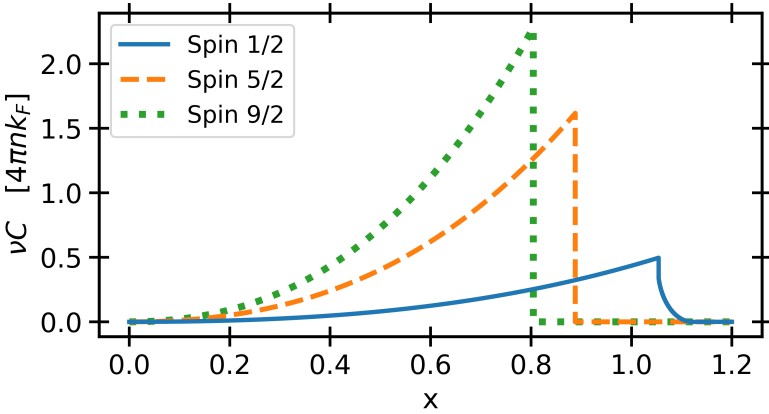

Figure 8: Tan's contact in SU(N) Fermi gases. $\nu C$ as a function of the gas parameter for three different spins: $s = 1/2$, $5/2$, and $9/2$. We use the second-order model for the three cases.

$P < 1$ (see Fig. 10). The behavior of the critical gas parameter values with the spin, in Fig. 10, can be understood by the balance between Fermi kinetic energy and interaction energy. Increasing the spin degeneracy originates in turn an increase in the number of interacting pairs of particles because the interatomic potential acts only between pairs of different $z$-spin component. In other words, the Fermi gas becomes more interacting and reaches the fully polarized (ferromagnetic) phase at lower $x$ values.

## 5 Conclusions

Summarizing, we present the analytic expression of the energy of a repulsive SU(N) Fermi gas, in terms of the spin-channel occupations, at second order of the gas parameter. This analytic derivation allows for an accurate estimation of the magnetic properties of the Fermi gas for any value of the spin. Moreover, by using the analytical solution one directly avoids any uncertainty coming from the numerical integration. This is in fact quite important since the function to integrate in the second-order term has many singular points and, if one does not apply a previous analytical treatment to the function, the numerical method may simply diverge. In

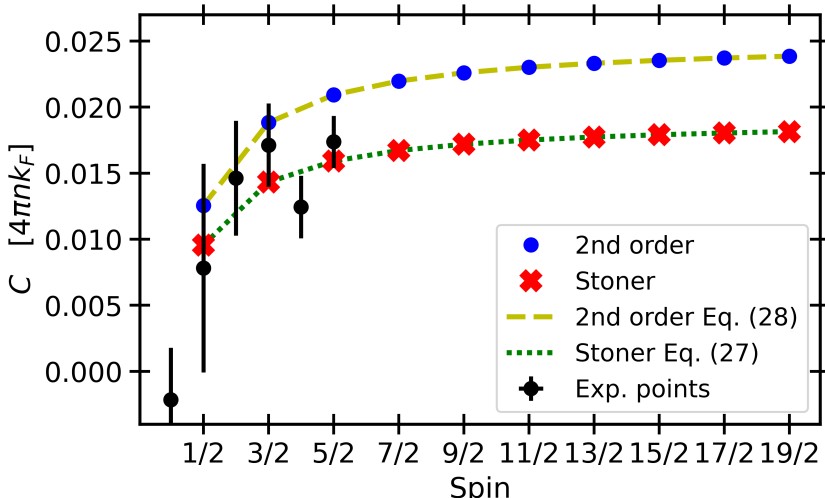

Figure 9: Tan's contact in SU(N) Fermi gases. Evolution of $C$ with the spin of particles at fixed density $x = 0.3$ for the Stoner model and the second order model; the points with error bars are experimental data from Ref. [36] re-scaled by a factor of $(2\pi)^2$. The green and yellow lines follow the behavior of the Tan's contact of Eq. (27) and Eq. (28) respectively.

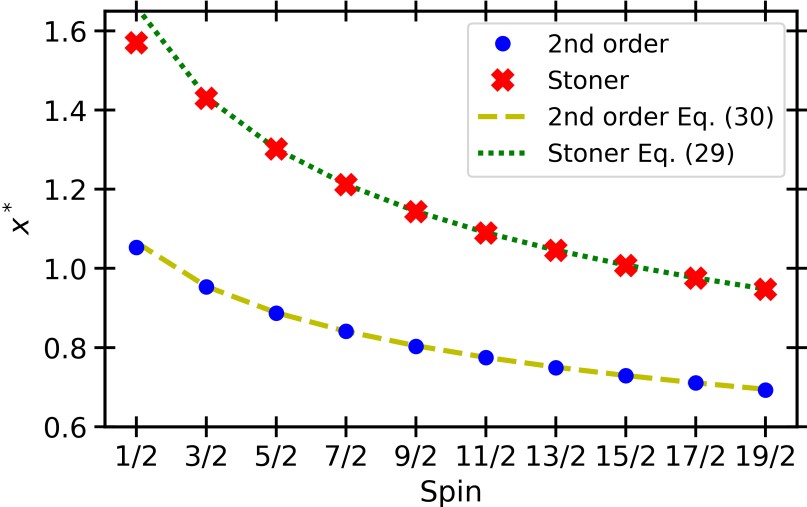

Figure 10: Critical gas parameter values of the ferromagnetic transition as a function of the spin for the Stoner model and the second order model. The green and yellow lines follow the behavior predicted by Eqs. (29) and (30).

order to study the system, we have chosen the occupational configuration and the polarization protocol that minimizes the energy: at $P = 0$, all the species are equally occupied; and, when we increase $P$, one species increases and all the rest diminish in the same manner. However, we point out that our formalism can be applied to any occupational configuration and any polarization protocol. One just needs to find the new expressions for the fractions of $\lambda$ particles $C_\lambda$. In fact, in many experiments, the species' occupation can be tuned or controlled almost at will [24, 26] and there are studies that have dealt with imbalanced systems [25]. However, we point out that all these other configurations are excited states, they have more energy than the configuration we have chosen. Due to that, if a Fermi system can thermalise, it will fall to

the behavior we have predicted in this work. Using another occupational configuration and a different polarization protocol leads to different critical values of the gas parameter $x^*$.

Our results show that the ferromagnetic transition turns out to be first-order for $s = 1/2$, in contrast with the continuous transition obtained at the Hartree-Fock (Stoner) approximation [2]. At second order, and for $s > 1/2$, the phase transition is always of first-order type. Our derivation applies to any spin of the Fermi particles and this allows for the study of itinerant ferromagnetism in SU(N) fermions. The critical gas parameter for spin $1/2$ decreases significantly with respect to the Stoner model, approaching the experimental estimation $x \simeq 1$ [7]. Remarkably, the critical $x$ decreases monotonically with the spin of the particles suggesting that the observation of itinerant ferromagnetism could be favored by working with highly-degenerate gases as Yb [28] and Sr [29]. In order to verify experimentally our predictions, one could use a similar method to the one used in Ref. [7] where the Fermi system can thermalise, but generalized to larger spin degeneracies.

Beyond the second-order terms analyzed in this work, the energy ceases to be universal in terms of the gas parameters because other scattering parameters of the interactions, mainly the $s$-wave effective range and $p$-wave scattering length get involved [30]. However, these corrections are expected to be small due to the diluteness of the Fermi gases in experiments that make scattering be dominated by $s$-wave scattering.

## Acknowledgments

We acknowledge financial support from MCIN/AEI/10.13039/501100011033 (Spain) grant No. PID2020-113565GB-C21 and from Secretaria d'Universitats i Recerca del Departament d'Empresa i Coneixement de la Generalitat de Catalunya, co-funded by the European Union Regional Development Fund within the ERDF Operational Program of Catalunya (project QuantumCat, ref. 001-P-001644).

## A  Polarization in SU(N) Fermi gases

We have considered that when the occupation of one of the spin channels grows, the rest decreases in equal form, keeping the total number of particles as constant. This results from the fact that the Hamiltonian does not depend on the angular momentum of each species, hence, all the species are equivalent. Moreover, this fact can be proved by minimizing the energy,

$$E = \frac{3}{5} N \epsilon_F \frac{1}{\nu} \sum_{\lambda} C_{\lambda}^{5/3} + V(C_s, \ldots, C_{-s}), \tag{A.1}$$

with $\sum_{\lambda} C_{\lambda} = \nu$. We define the channel that grows,

$$C_+ = \nu - \sum_{\lambda \neq +} C_{\lambda}, \tag{A.2}$$

and substitute it in the energy,

$$E = \frac{3}{5} N \epsilon_F \frac{1}{\nu} \left[ \left( \nu - \sum_{\lambda \neq +} C_{\lambda} \right)^{5/3} + \sum_{\lambda \neq +} C_{\lambda}^{5/3} \right] + V\left( \nu - \sum_{\lambda \neq +} C_{\lambda}, C_{s-1} \ldots, C_{-s} \right). \tag{A.3}$$

Now, we minimize the energy for all the channels, except for the one that we have isolated which, due to the constraint (A.2), depends on the others,

$$\frac{\partial E}{\partial C_{\sigma}} = 0 \quad \forall \sigma \neq +. \tag{A.4}$$

Then,

$$\frac{\partial E}{\partial C_\sigma} = N\epsilon_F \frac{1}{\nu}\left[ C_\sigma^{2/3} - \left(\nu - \sum_{\lambda \neq +} C_\lambda\right)^{2/3} \right] + \frac{\partial V}{\partial C_\sigma} - \frac{\partial V}{\partial C_+} = 0\,. \tag{A.5}$$

Rearranging terms, we obtain

$$N\epsilon_F \frac{1}{\nu} C_\sigma^{2/3} + \frac{\partial V}{\partial C_\sigma} = N\epsilon_F \frac{1}{\nu}\left(\nu - \sum_{\lambda \neq +} C_\lambda\right)^{2/3} + \frac{\partial V}{\partial C_+}\,. \tag{A.6}$$

From Eq. (A.6), one can see that all the decreasing $C_\sigma$ satisfy the same equation, therefore, they must be equal.

# B  Parameters of the Landau expansion

In this section we show the expression for all the parameters of the Landau expansion in terms of the gas degeneracy ($\nu = 2s + 1$) and the gas parameter ($x = k_F a_0$). We first show the Landau expansion up to fourth order in $P$ including the logarithmic term that comes from the second order term in perturbation theory.

$$f - f_0 = -\frac{A}{2}(\bar{x} - x_0)P^2 - \frac{B}{3}|P|^3 + \frac{C}{4}P^4 + \frac{L}{4}P^4 \ln|P|\,. \tag{B.1}$$

The parameters that are invariant in any order are

$$f = \frac{5E}{3N\epsilon_F}\,, \quad A = \frac{20}{9\pi}(\nu - 1)\,, \quad x_0 = \pi/2\,. \tag{B.2}$$

We split this section into two subsections. The first one contains the expression of the parameters up to first order in perturbation theory (Hartree-Fock), and the second one up to second order.

## B.1  1st order parameters

$$\begin{aligned}
&f_0 = 1 + \frac{10}{9\pi}(\nu - 1)x\,, \quad \bar{x} = x\,, \quad B = \frac{5}{27}(\nu - 1)(\nu - 2)\,, \\
&C = \frac{20}{243}(\nu - 1)(\nu^2 - 3\nu + 3)\,, \quad L = 0\,.
\end{aligned} \tag{B.3}$$

## B.2  2nd order parameters

$$\begin{aligned}
f_0 &= 1 + \frac{10}{9\pi}(\nu - 1)x + \frac{4}{21\pi^2}(\nu - 1)(11 - 2\ln 2)x^2\,, \\
\bar{x} &= x + \frac{2}{15\pi}(22 - 7\nu + 4(\nu - 1)\ln 2)x^2\,, \\
B &= \frac{5}{27}(\nu - 1)(\nu - 2) + \frac{2}{81\pi^2}(\nu - 1)(\nu - 2)(44 + \nu + 8(\nu - 1)\ln 2)x^2\,, \\
C &= \frac{20}{243}(\nu - 1)(\nu^2 - 3\nu + 3) \\
&\quad + (\nu - 1)\frac{528 - 336\nu - 16\nu^2 + 29\nu^3 - (96 - 192\nu + 128\nu^2 - 32\nu^3)\ln 2}{729\pi^2}x^2 \\
&\quad + \frac{20\nu^3}{243\pi^2}(\nu - 1)\ln\frac{\nu}{6}x^2\,, \\
L &= \frac{20\nu^3}{243\pi^2}(\nu - 1)x^2\,.
\end{aligned} \tag{B.4}$$

For spin 1/2, these expressions reduce to

$$f_0 = 1 + \frac{10}{9\pi}x + \frac{4(\nu-1)(11-2\ln 2)}{21\pi^2}x^2, \qquad \overline{x} = x + \frac{8(2+\ln 2)}{15\pi}x^2, \qquad B = 0,$$

$$C = \frac{20}{243} + \frac{8(3+4\ln 2)}{729\pi^2}x^2 - \frac{160\ln 3}{243\pi^2}x^2, \qquad L = \frac{160}{243\pi^2}x^2 .$$

(B.5)

## C Calculation of the second-order term

In the following, we detail the calculation of the second-order term for the energy,

$$\frac{E}{N} = \epsilon_F \frac{1}{\nu k_F^5} \sum_{\lambda_1,\lambda_2} I_2(k_{F,\lambda_1}, k_{F,\lambda_2}) a_0^2 (1 - \delta_{\lambda_1,\lambda_2}), \quad \text{where}$$

(C.1)

$$I_2(k_{F,\lambda_1}, k_{F,\lambda_2}) = \frac{3}{16\pi^5} \int d\mathbf{l}\, n_l \int d\mathbf{k}\, n_k \int 2d\mathbf{q} d\mathbf{q}' \frac{1-(1-n_q)(1-n_{q'})}{q^2+q'^2-k^2-l^2} \delta(\mathbf{q}+\mathbf{q}'-\mathbf{k}-\mathbf{l}).$$

We need to calculate the multiple integral $I_2$. The first step is to expand the inner part, where the occupation functions appear. One, then, obtains an expression with $n_q + n_{q'} - n_q n_{q'}$. The terms containing $n_q$ and $n_{q'}$ correspond to integrals of spheres (SI) with respect to $\mathbf{q}$ or $\mathbf{q}'$, which can be integrated. The rest of the integrals with respect to $\mathbf{k}$ and $\mathbf{l}$ are quite arduous to integrate, but after some lengthy calculations, one can obtain them. And finally, the term proportional to $n_q n_{q'}$, corresponding to the volume of the intersection of two spheres (SII) is zero due to symmetry reasons.

We rewrite the expression of $I_2$ using the integrated functions SI and SII,

$$I_2(k_{F,\lambda_1}, k_{F,\lambda_2}) = \frac{3}{16\pi^5} \int d\mathbf{l}\, n_l \int d\mathbf{k}\, n_k \int 2d\mathbf{q} d\mathbf{q}' \frac{n_q + n_{q'} - n_q n_{q'}}{q^2+q'^2-k^2-l^2} \delta(\mathbf{q}+\mathbf{q}'-\mathbf{k}-\mathbf{l})$$

(C.2)

$$= \frac{3}{16\pi^5} \int d\mathbf{l}\, n_l \int d\mathbf{k}\, n_k \left[ SI(P,R,k_{F,\lambda_1}) + SI(P,R,k_{F,\lambda_2}) - SII(P,R,k_{F,\lambda_1}, k_{F,\lambda_2}) \right].$$

First of all, let's show that the SII term is zero. In order to follow the derivation, we need to have in mind that the momenta $\mathbf{k}$ and $\mathbf{q}$ run over the same momentum $k_{F,\lambda_1}$, and that the momenta $\mathbf{l}$ and $\mathbf{q}'$ do so over $k_{F,\lambda_2}$. The procedure is the following. We split the integral in identical parts. The first part is integrated with respect to $\mathbf{l}$, the second one with respect to $\mathbf{q}'$, both running over the same values. We note that it could have been done integrating with respect to $\mathbf{k}$ and $\mathbf{q}$, instead of $\mathbf{l}$ and $\mathbf{q}'$. Then, we slightly manipulate the two expressions and we obtain two identical integrals but with opposite signs, hence, they cancel each other,

$$\int d\mathbf{l}\, n_l \int d\mathbf{k}\, n_k \int 2d\mathbf{q} d\mathbf{q}' \frac{n_q n_{q'}}{q^2+q'^2-k^2-l^2} \delta(\mathbf{q}+\mathbf{q}'-\mathbf{k}-\mathbf{l})$$

$$= 2 \int d\mathbf{l} d\mathbf{k} d\mathbf{q} d\mathbf{q}' \frac{n_l n_k n_q n_{q'}}{q^2+q'^2-k^2-l^2} \delta(\mathbf{q}+\mathbf{q}'-\mathbf{k}-\mathbf{l})$$

$$= \int d\mathbf{l} d\mathbf{k} d\mathbf{q} d\mathbf{q}' \frac{n_l n_k n_q n_{q'}}{q^2+q'^2-k^2-l^2} \delta(\mathbf{q}+\mathbf{q}'-\mathbf{k}-\mathbf{l})$$

(C.3)

$$+ \int d\mathbf{l} d\mathbf{k} d\mathbf{q} d\mathbf{q}' \frac{n_l n_k n_q n_{q'}}{q^2+q'^2-k^2-l^2} \delta(\mathbf{q}+\mathbf{q}'-\mathbf{k}-\mathbf{l})$$

$$= \int d\mathbf{k} d\mathbf{q} d\mathbf{q}' \frac{n_k n_q n_{q'}}{-2k^2+2\mathbf{k}(\mathbf{q}+\mathbf{q}')-2\mathbf{q}\mathbf{q}'} + \int d\mathbf{l} d\mathbf{k} d\mathbf{q} \frac{n_l n_k n_q}{2q^2-2\mathbf{q}(\mathbf{k}+\mathbf{l})+2\mathbf{k}\mathbf{l}}$$

$$= -\frac{1}{2} \int d\mathbf{k} d\mathbf{q} d\mathbf{q}' \frac{n_k n_q n_{q'}}{k^2-\mathbf{k}(\mathbf{q}+\mathbf{q}')+\mathbf{q}\mathbf{q}'} + \frac{1}{2} \int d\mathbf{q} d\mathbf{k} d\mathbf{l} \frac{n_q n_k n_l}{q^2-\mathbf{q}(\mathbf{k}+\mathbf{l})+\mathbf{k}\mathbf{l}} = 0.$$

With this, $I_2$ becomes

$$I_2(k_{F,\lambda_1}, k_{F,\lambda_2}) = \frac{3}{16\pi^5} \int d\mathbf{l} n_l \int d\mathbf{k} n_k \left[ SI(P,R,k_{F,\lambda_1}) + SI(P,R,k_{F,\lambda_2}) \right]. \qquad (C.4)$$

We will integrate only one SI; the other one will be the same but interchanging the Fermi momenta. In the end, we will add both expressions. The inner part of the integral is

$$
\begin{aligned}
SI(k,l,k_{F,\lambda}) &= \int 2d\mathbf{q} d\mathbf{q}' \frac{n_q}{q^2 + q'^2 - k^2 - l^2} \delta(\mathbf{q} + \mathbf{q}' - \mathbf{k} - \mathbf{l}) \\
&= \int d\mathbf{q} \frac{n_q}{q^2 - \mathbf{q} \cdot (\mathbf{k} + \mathbf{l}) + \mathbf{k} \cdot \mathbf{l}} \\
&= \int_0^{k_{F,\lambda}} q^2 dq \int_0^\pi \sin\theta \, d\theta \frac{2\pi}{q^2 - q|\mathbf{k}+\mathbf{l}| + \mathbf{k} \cdot \mathbf{l}} \\
&= 2\pi \int_0^{k_{F,\lambda}} q^2 dq \frac{1}{q|\mathbf{k}+\mathbf{l}|} \ln\left| \frac{q^2 + q|\mathbf{k}+\mathbf{l}| + \mathbf{k} \cdot \mathbf{l}}{q^2 - q|\mathbf{k}+\mathbf{l}| + \mathbf{k} \cdot \mathbf{l}} \right| \\
&= \frac{2\pi}{|\mathbf{k}+\mathbf{l}|} \left\{ \left( \frac{k_{F,\lambda}^2}{2} - \frac{k^2 + l^2}{4} \right) \ln\left| \frac{k_{F,\lambda}^2 + k_{F,\lambda}|\mathbf{k}+\mathbf{l}| + \mathbf{k} \cdot \mathbf{l}}{k_{F,\lambda}^2 - k_{F,\lambda}|\mathbf{k}+\mathbf{l}| + \mathbf{k} \cdot \mathbf{l}} \right| - \right. \\
&\quad \left. \frac{|\mathbf{k}+\mathbf{l}||\mathbf{k}-\mathbf{l}|}{4} \ln\left| \frac{k_{F,\lambda}^2 + k_{F,\lambda}|\mathbf{k}-\mathbf{l}| - \mathbf{k} \cdot \mathbf{l}}{k_{F,\lambda}^2 - k_{F,\lambda}|\mathbf{k}-\mathbf{l}| - \mathbf{k} \cdot \mathbf{l}} \right| + k_{F,\lambda}|\mathbf{k}+\mathbf{l}| \right\}.
\end{aligned}
\qquad (C.5)
$$

We write everything in terms of the modules of $\mathbf{k}$ and $\mathbf{l}$ and the angle between them. In the angular part, we change the variable to $x = -\cos\theta$,

$$
\begin{aligned}
|\mathbf{k}+\mathbf{l}| &= \sqrt{k^2 + l^2 + 2kl\cos\theta} = \sqrt{k^2 + l^2 - 2klx}, \\
|\mathbf{k}-\mathbf{l}| &= \sqrt{k^2 + l^2 - 2kl\cos\theta} = \sqrt{k^2 + l^2 + 2klx}, \\
\mathbf{k} \cdot \mathbf{l} &= kl\cos\theta = -klx, \\
\int d\mathbf{l} n_l \int d\mathbf{k} n_k &= 2(2\pi)^2 \int_0^{k_{F,\lambda_1}} k^2 dk \int_0^{k_{F,\lambda_2}} l^2 dl \int_0^\pi \sin\theta \, d\theta \\
&= 2(2\pi)^2 \int_0^{k_{F,\lambda_1}} k^2 dk \int_0^{k_{F,\lambda_2}} l^2 dl \int_{-1}^1 dx.
\end{aligned}
\qquad (C.6)
$$

We take out the $2\pi$ from SI and integrate over x,

$$
\begin{aligned}
\frac{1}{2\pi} \int dx SI(k,l,q) &= \frac{2}{3}qx + \frac{1}{kl}\left( \frac{k^2}{4} + \frac{l^2}{4} - \frac{q^2}{2} \right) \sqrt{k^2 + l^2 - 2klx} \ln\left| \frac{q^2 + q\sqrt{k^2 + l^2 - 2klx} - klx}{q^2 - q\sqrt{k^2 + l^2 - 2klx} - klx} \right| \\
&\quad - \frac{1}{12kl}(k^2 + l^2 + 2klx)^{3/2} \ln\left| \frac{q^2 + q\sqrt{k^2 + l^2 + 2klx} + klx}{q^2 - q\sqrt{k^2 + l^2 + 2klx} + klx} \right| \\
&\quad + \frac{k^4 + l^4 + 2k^2l^2 - 2k^2q^2 - 2l^2q^2 + q^4}{3kl\sqrt{k^2 + l^2 - q^2}} \ln\left| \frac{q\sqrt{k^2 + l^2 - q^2} + klx}{q\sqrt{k^2 + l^2 - q^2} - klx} \right| \\
&\quad - \frac{q^3}{3kl}\left( \ln\left| q\sqrt{k^2 + l^2 - q^2} + klx \right| + \ln\left| q\sqrt{k^2 + l^2 - q^2} - klx \right| \right). \quad (C.7)
\end{aligned}
$$

The first external integral is

$$
\begin{aligned}
SX(k,l,q) &= \frac{1}{2\pi}\int_{-1}^{1}dx\,SI(k,l,q)\\
&= \frac{4}{3}qx + \frac{(k-l)(2k^2+2l^2-kl-3q^2)}{6kl}\ln\left|\frac{q^2+q(k-l)-kl}{q^2-q(k-l)-kl}\right|\\
&\quad - \frac{(k+l)(2k^2+2l^2+kl-3q^2)}{6kl}\ln\left|\frac{q^2+q(k+l)+kl}{q^2-q(k+l)+kl}\right|\\
&\quad + \frac{4}{6kl}(k^2+l^2-q^2)^{3/2}\ln\left|\frac{q\sqrt{k^2+l^2-q^2}+kl}{q\sqrt{k^2+l^2-q^2}-kl}\right|.
\end{aligned}
\tag{C.8}
$$

The next step is the integration over $k$ and $l$. We will proceed in two ways (both producing the same result):

1. $\int_0^k dk \int_0^q dl\, k^2 l^2 SX(k,l,q)$,

2. $\lim_{k\to q}\int_0^k dk \int_0^l dl\, k^2 l^2 SX(k,l,q)$.

1)

$$
\begin{aligned}
\int dl\, k^2 l^2 SX(k,l,q) &= \frac{k}{120}\Bigl[4klq(-4k^2+7q^2)+44l^3klq\\
&\quad + (16k^5-40k^3q^2+30kq^4+40k^3l^2+30kl^4-60kl^2q^2)\ln\left|\frac{l-q}{l+q}\right|\\
&\quad + (16l^5+40l^3(k-q)(k+q))\ln\left|\frac{k-q}{k+q}\right|\\
&\quad + 16(k^2+l^2-q^2)^{5/2}\ln\left|\frac{q\sqrt{k^2+l^2-q^2}+kl}{q\sqrt{k^2+l^2-q^2}-kl}\right|\Bigr].
\end{aligned}
\tag{C.9}
$$

It can be easily checked that the function (C.9) is zero when $l=0$. Also, we will need to make use of the following limit,

$$
\lim_{l\to q}\ln\left|\frac{q\sqrt{k^2+l^2-q^2}+kl}{q\sqrt{k^2+l^2-q^2}-kl}\right| = -\lim_{l\to q}\ln\left|\frac{l-q}{l+q}\right| - \ln\left|\frac{k-q}{k+q}\right| - 2\ln\left|\frac{k+q}{k}\right|.
\tag{C.10}
$$

The definite integral becomes

$$
\begin{aligned}
\int_0^q dl\, k^2 l^2 SX(k,l,q) &= \frac{k}{120}\Bigl[-16k^3q^2+72kq^4+(16q^5+40q^3(k-q)(k+q)\\
&\quad -16k^5)\ln\left|\frac{k-q}{k+q}\right| - 32k^5\ln\left|\frac{k+q}{k}\right|\Bigr].
\end{aligned}
\tag{C.11}
$$

Finally, the integral over $k$ is

$$
\begin{aligned}
\int_0^k dk\int_0^q dl\, k^2 l^2 SX(k,l,q) &= \frac{1}{420}\Bigl[-8k^5q^2+66k^3q^4+30kq^6\\
&\quad + (-8k^7+35k^4q^3-42k^2q^5+15q^7)\ln\left|\frac{k-q}{k+q}\right|\\
&\quad - 16k^7\ln\left|\frac{k+q}{k}\right|\Bigr].
\end{aligned}
\tag{C.12}
$$

2)

$$\int_0^k dk \int_0^l dl\, k^2 l^2 SX(k,l,q) = \frac{1}{840}\Bigg[-16k^5lq - 16kl^5q + 88k^3l^3q + 44k^3lq^3 + 44kl^3q^3 + 32klq^5$$

$$+ (16l^7 - 56l^5q^2 + 70l^3q^4 + 70k^4l^3 + 56k^2l^5 - 140k^2l^3q^2)\ln\left|\frac{k-q}{k+q}\right|$$

$$+ (16k^7 - 56k^5q^2 + 70k^3q^4 + 70k^3l^4 + 56k^5l^2 - 140k^3l^2q^2)\ln\left|\frac{l-q}{l+q}\right|$$

$$+ 16(k^2+l^2-q^2)^{7/2}\ln\left|\frac{q\sqrt{k^2+l^2-q^2}+kl}{q\sqrt{k^2+l^2-q^2}-kl}\right|\Bigg]. \tag{C.13}$$

As the expression above is symmetrical with respect to $k$ and $l$, it does not matter which variable we choose to perform the limit to $(k,l) \to q$ since both results are formally equivalent. As we want to check that this second method gives the same expression as the one found in method 1, we will set the limit $l$ going to $q$,

$$\lim_{l\to q}\int_0^k dk \int_0^l dl\, k^2 l^2 SX(k,l,q) = \frac{1}{840}\Bigg[-16k^5q^2 + 132k^3q^4 + 60kq^6$$

$$+ (-16k^7 + 70k^4q^3 - 84k^2q^5 + 30q^7)\ln\left|\frac{k-q}{k+q}\right| \tag{C.14}$$

$$- 32k^7\ln\left|\frac{k+q}{k}\right|\Bigg].$$

We recover indeed the same expression.

Coming back to the integral we had at the beginning,

$$I_2(k_{F,\lambda_1}, k_{F,\lambda_2}) = \frac{3}{16\pi^5}\int dl\, n_l \int d\mathbf{k}\, n_k \Big[SI(P,R,k_{F,\lambda_1}) + SI(P,R,k_{F,\lambda_2})\Big]. \tag{C.15}$$

We substitute the results we have obtained for the integrals (Eqs. (C.12) or (C.14)) and we recover the factor $2(2\pi)^3$ coming from the angular integrals.

$$I_2 = \frac{3}{16\pi^5} 2(2\pi)^3 \Bigg(\lim_{k\to q}\int_0^k dk \int_0^l dl\, k^2 l^2 SX(k,l,q) + \lim_{l\to q'}\int_0^k dk \int_0^l dl\, k^2 l^2 SX(k,l,q')\Bigg)$$

$$= \frac{3}{16\pi^5}\frac{2(2\pi)^3}{420}\Bigg[-8k^5l^2 - 8l^5k^2 + 66k^3l^4 + 66l^3k^4 + 30kl^6 + 30lk^6$$

$$+ (7k^7 + 7l^7 + 35k^4l^3 + 35l^4k^3 - 42k^2l^5 - 42l^2k^5)\ln\left|\frac{k-l}{k+l}\right|$$

$$- 16k^7\ln\left|\frac{k+l}{k}\right| - 16l^7\ln\left|\frac{k+l}{l}\right|\Bigg]$$

$$= \frac{1}{140\pi^2}\Bigg[2kl(k+l)(15k^4 - 19k^3l + 52k^2l^2 - 19kl^3 + 15l^4)$$

$$+ 7(k+l)(k-l)^4(k^2+3kl+l^2)\ln\left|\frac{k-l}{k+l}\right| - 16\Big(k^7\ln\left|\frac{k+l}{k}\right| + l^7\ln\left|\frac{k+l}{l}\right|\Big)\Bigg]. \tag{C.16}$$

After putting everything together and rearranging terms, $I_2$ is written in terms of k and l as

$$I_2 = \frac{4}{35\pi^2}\Bigg[\frac{1}{8}kl(k+l)(15k^4 - 19k^3l + 52k^2l^2 - 19kl^3 + 15l^4)$$

$$+ \frac{7}{16}(k+l)(k-l)^4(k^2+3kl+l^2)\ln\left|\frac{k-l}{k+l}\right| - \Big(k^7\ln\left|\frac{k+l}{k}\right| + l^7\ln\left|\frac{k+l}{l}\right|\Big)\Bigg]. \tag{C.17}$$

Replacing $k$ and $l$ by the Fermi momenta ($k_{F,\lambda_1}$ and $k_{F,\lambda_2}$),

$$
\begin{aligned}
I_2 = \frac{4}{35\pi^2}\Big[ &\frac{1}{8}k_{F,\lambda_1}k_{F,\lambda_2}(k_{F,\lambda_1}+k_{F,\lambda_2})(15k_{F,\lambda_1}^4 - 19k_{F,\lambda_1}^3 k_{F,\lambda_2}+52k_{F,\lambda_1}^2 k_{F,\lambda_2}^2 - 19k_{F,\lambda_1}k_{F,\lambda_2}^3 \\
&+15k_{F,\lambda_2}^4) + \frac{7}{16}(k_{F,\lambda_1}+k_{F,\lambda_2})(k_{F,\lambda_1}-k_{F,\lambda_2})^4(k_{F,\lambda_1}^2 + 3k_{F,\lambda_1}k_{F,\lambda_2} + k_{F,\lambda_2}^2)\ln\left|\frac{k_{F,\lambda_1}-k_{F,\lambda_2}}{k_{F,\lambda_1}+k_{F,\lambda_2}}\right| \\
&-\Big(k_{F,\lambda_1}^7\ln\left|\frac{k_{F,\lambda_1}+k_{F,\lambda_2}}{k_{F,\lambda_1}}\right| + k_{F,\lambda_2}^7\ln\left|\frac{k_{F,\lambda_1}+k_{F,\lambda_2}}{k_{F,\lambda_2}}\right|\Big)\Big].
\end{aligned}
\tag{C.18}
$$

Now, we replace the Fermi momenta by $k_F C_\lambda^{1/3}$,

$$
\begin{aligned}
I_2 = \frac{4k_F^7}{35\pi^2}\Big[ &\frac{1}{8}C_{\lambda_1}^{1/3}C_{\lambda_2}^{1/3}\big(C_{\lambda_1}^{1/3}+C_{\lambda_2}^{1/3}\big)\big(15C_{\lambda_1}^{4/3}-19C_{\lambda_1}C_{\lambda_2}^{1/3}+52C_{\lambda_1}^{2/3}C_{\lambda_2}^{2/3}-19C_{\lambda_1}^{1/3}C_{\lambda_2}+15C_{\lambda_2}^{4/3}\big) \\
&+\frac{7}{16}\big(C_{\lambda_1}^{1/3}+C_{\lambda_2}^{1/3}\big)\big(C_{\lambda_1}^{1/3}-C_{\lambda_2}^{1/3}\big)^4\big(C_{\lambda_1}^{2/3}+3C_{\lambda_1}^{1/3}C_{\lambda_2}^{1/3}+C_{\lambda_2}^{2/3}\big)\ln\left|\frac{C_{\lambda_1}^{1/3}-C_{\lambda_2}^{1/3}}{C_{\lambda_1}^{1/3}+C_{\lambda_2}^{1/3}}\right| \\
&-\Big(C_{\lambda_1}^{7/3}\ln\left|\frac{C_{\lambda_1}^{1/3}+C_{\lambda_2}^{1/3}}{C_{\lambda_1}^{1/3}}\right| + C_{\lambda_2}^{7/3}\ln\left|\frac{C_{\lambda_1}^{1/3}+C_{\lambda_2}^{1/3}}{C_{\lambda_2}^{1/3}}\right|\Big)\Big].
\end{aligned}
\tag{C.19}
$$

Finally, in terms of $y = (C_{\lambda_1}/C_{\lambda_2})^{1/3}$,

$$
\begin{aligned}
I_2 = \frac{4k_F^7}{35\pi^2}C_{\lambda_1}C_{\lambda_2}\frac{C_{\lambda_1}^{1/3}+C_{\lambda_2}^{1/3}}{2}\Big[ &\frac{1}{4}\big(15y^2-19y+52-19y^{-1}+15y^{-2}\big) \\
&+\frac{7}{8}y^{-2}(y-1)^4(y+3+y^{-1})\ln\left|\frac{1-y}{1+y}\right| \\
&-\frac{2y^4}{1+y}\ln\left|1+y^{-1}\right| - \frac{2y^{-4}}{1+y^{-1}}\ln\left|1+y\right|\Big].
\end{aligned}
\tag{C.20}
$$

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
