# Peer review of "Itinerant ferromagnetism in dilute SU(N) Fermi gases"

_SciPost Physics, doi:SciPost Phys. 14, 038 (2023)_

## Round 2 · Referee Report · Anonymous (Referee 1) · 2022-6-28

Strengths

1- The manuscript derives analytic expressions for the energy of repulsive Fermi gases up to 2nd order of perturbation theory in the gas parameter. Analytic expressions are always useful for guidance and comparison with experiments, where the approximations are valid.

2- On Fig. 9 the authors compare their results against experimental data, which is well described by their theory. This is a good indicator of its validity and predictive nature.

Weaknesses

1 - The current manuscript fails to provide a complete review of the literature, presenting results from computational and experimental teams regarding the physics of SU(N) fermions. I provide a series of articles in my report.

2- Some interesting physics arises (such as long-lived prethermal states) from initial conditions where there is a spin imbalance in for the different spin projections, as opposed to the case considered in the manuscript.

Report

The authors present a theoretical study of itinerant ferromagnetism up to second order in the gas parameter, presenting analytic expressions which correct the Stoner results, i.e. corrections beyond the Hartree-Fock calculation.

Authors analyze how the results depend on spin degree of freedom, comparing results against $s= \frac{1}{2}$ up to $s=\frac{9}{2}$. The main conclusions of the paper are: a) The nature of the transition is first-order, b) the critical density reduces as the total spin decreases, and c) the character of the ferromagnetic phase transition changes from being continuous to first order in case of $s=\frac{1}{2}$ when the second order correction in $x$ is included.

Authors claim these results can provide a tool for the exploration of itinerant ferromagnetism with alkaline-earth-like atoms (AEAs). I can recommend publication in SciPost Phys after the following revision is completed.

$\mathbf{Major \, comments}$ 1) My main concern is the lack of a discussion on the convergence of the results. It is known that perturbative calculations are not necessarily convergent. Given that the results for $s=\frac{1}{2}$ exhibit the most dramatic change (the nature of the ferromagnetic transition), this raises questions on the convergence of the calculations. A possible way of addressing this would be to compare the Hartree-Fock solution against their second order results in a controlled $1/N$ expansion, where the effects of order $\mathcal{O}(x^2)$ and the effects of large-$N$ can be explored (where $N=2S+1$).

2) Section III needs restructuring. On a first inspection, eqs. (14) and (15) are misleading, since it seems that although authors perform a calculation of the energy up to second order in the gas parameter $x$ they only compute the free energy to first order in $x$, i.e. the Hartree-Fock (which they use to determine the nature of the transition). This would be an uncontrolled way of performing perturbation theory and therefore puts under question the validity of the findings. One needs to go to the appendix to realize that the different terms are of order $\mathcal{O}(x^2)$. This is however not clear from the manuscript and needs to be clarified early on in section III.

3) In the introduction the authors should briefly discuss the progress with ultracold atoms and SU(N) physics with AEAs in the last years. Here I present some suggestions:
a) PRResearch 2, 012028(R) (2020) studied the collective excitations in SU(N) Fermi gases. b) PRL 121, 167205 (2018) studied interaction effects in SU(N) Fermi gases as N is varied. c) PRA 104, 043316 (2021) found universal behavior of thermodynamic quantities with N in SU(N) gases in optical lattices. d) PRA 101, 053620 (2020) studied prethermalization in SU(N) gases where different initial conditions are studied before a quench. e) Nat. Phys. 10, 779 (2014) was one of the first studies to demonstrate the SU(N) symmetric interactions in AEAs. f) arxiv:2010.07730 observed antiferromagnetic correlations in SU(N) gases in optical lattices and achieved extremely low temperatures. g) Nat. Phys. 16, 1216 (2020) perform a thorough study of the thermodynamics of deeply degenerate SU(N) Fermi gases.

4) From Fig. 4 is unclear that the Stoner model predicts a diverging $\chi$. I suggest making this explicit from eq. (22).

5) Figure 5 is present in the manuscript, but is never discussed in the manuscript. Either is not relevant and it should be deleted or proper discussion should be presented.

6) On page 5, first column authors mention: ' In both cases, the polarization jumps from $P=0$ to $1$ without intermediate values, in contrast to the case of $s=\frac{1}{2}$'. This sentence is slightly unclear. Authors should mention that is the minimum of the energy as a function of the polarization what jumps. For example: ' In both cases, the location of the minimum of the energy as a function of polarization jumps from $P=0$ to $1$ without intermediate values, in contrast to the case of $s=\frac{1}{2}$'.

7) On page 6, authors mention 'In Fig. 9, we also plot experimental results from Ref. [27] that show a similar behavior to the one observed with our second order expansion (once properly rescaled)'. Authors should give a brief explanation of how this rescaling was done.

8) For large-spin, some interesting physics arises (such as long-lived prethermal states) from initial conditions where there is a spin imbalance in for the different spin projections [PRA 101, 053620 (2020)], as opposed to the particular polarization case considered in the manuscript. Can authors comment on how they expect their results to change when other polarizations are considered?

$\mathbf{Minor \, comments}$ 1) Reference [7] Valtolina et. al., Nat Phys 13 704 (2017) was performed with $^6$Li atoms, which is the fermionic isotope.

2) I strongly advice the use of visually perceptive colormaps for Figures 3 and 6. The use of a visually perceptive colormap is not only useful if the paper is printed in black and white, but carries informational content. For example: In Figure 6 results of $E/N$ are presented as a function of $P$ for different values of $x$. Authors should use a visually perceptive colormap, since this would immediately indicate the reader which curve corresponds to smaller/larger value of $x$.

3) In Figure 3, what is $c$? Should it be $x$ instead?

4) On page4, last paragraph authors first mention Figure 7 and then Figure 6. They should change the order of the figures to ensure a coherent narrative.

5) In Figures 3 and 6, authors should use a markers to indicate the location of the minima for the different curves (helps with the discussion on the main text).

6) On Figure 10, have the authors tried fitting the data? Would be interesting to know how does $x$ scale with $N$. In addition, presenting the results from a large-$N$ expansion. Its validity can help diagnose where simple approximations are still valid and can simplify their analytic expressions.

7) In the conclusions, authors mention 'Beyond the second-order terms analyzed in this work, the energy ceases to be universal in terms of the gas parameters because other scattering parameters of the interactions, mainly the $s$-wave effective range and $p$-wave scattering length get involved.' Given that ultracold atom experiments deal with low-energy physics, mainly interactions are dominated by $s$-waves; authors should connect this to justify why neglecting higher-order terms and/or higher partial-waves scattering lengths is adequate.

$\mathbf{Typos \, and \, suggestions}$ 1) In the abstract: 'This is an extension of an old result' $\to$ 'This is an extension of previous results'.

2) On page 2, second column: 'We have been able to integrate Eq. (7) and find an...' $\to$ 'We have been able to integrate Eq. (7) and found an...' .

3) On page 3, section IV: 'formailsm' $\to$ 'formalism'.

4) On page 5, first column: '... we have access to other important properties as the Tan constant.' $\to$ '... we have access to other important properties such as the Tan contact.' (is Tan contact rather than Tan constant).

4) On page 6, first column: 'In other words, the Fermi gas becomes more interactive and...' $\to$ 'In other words, the Fermi gas becomes more interacting and...'

Requested changes

1- Include a literature review in the introduction.

2- For Figs. 3, 6, 8 use a perceptive color map.

3- For Figs. 3 and 6 also indicate with a marker the location of the minima (for easier visualization purposes).

4- Other requested changes are discussed in the report.

  • validity: good
  • significance: high
  • originality: good
  • clarity: good
  • formatting: reasonable
  • grammar: good

Author:  Jordi Boronat  on 2022-10-27  [id 2957]

(in reply to Report 1 on 2022-06-28)

Author: We kindly thank the Referee for his/her positive and extensive comments
and for finding
our manuscript appropriate for SciPost Phys. In the following, we address the
comments and
suggestions pointed out by the Referee and the changes introduced in the
manuscript accordingly.

Concern 1: My main concern is the lack of a discussion on the convergence
of the results. It is known that perturbative calculations are not necessarily
convergent. Given that the results for $s=1/2$ exhibit the most dramatic change
(the nature of the ferromagnetic transition), this raises questions on the
convergence of the calculations. A possible way of addressing this would be to
compare the Hartree-Fock solution against their second order results in a
controlled $1/N$ expansion, where the effects of
order $O(x^2)$ and the effects of large-$N$ can be explored (where $N=2S+1$).

Author:Our approach is perturbation theory in terms of the gas parameter $x$
and
a rigorous mathematical prove of convergence of the series is lacking. The
perturbative expansion of Fermi gases is only known (for zero polarization) up
to fourth order and no-one knows higher-order terms. In
principle, as the energy of the Fermi gas is finite, it should converge, at
least for low values of the gas parameter $x$. For perturbative expansions of
(dilute) Fermi gases, the regime of applicability is $x<1$ (after Eq. (10) we
have added a comment about this). The experimental Fermi gases in which we are
interested work precisely under this regime, hence, we expect to obtain a
better approximation to the true values when increasing the order. A numerical
comparison with QMC results for spin s=1/2 and zero polarization can be found
in Ref. [14]; there one can see that the series approaches QMC data when the
order of the perturbative series grows.

On the other hand, the behavior of the series when the degeneracy $\nu$ grows
was not explored in our first version. Following the advice of the Referee, we
have calculated the limit of our results for large spin degeneracy.
We have added in the end of Sec. II the limit $\nu\rightarrow\infty$ of Eq.
(10). This limit does not diverge and gives a finite expression.

Concern 2: Section III needs restructuring. On a first inspection, eqs.(14)
and (15) are misleading, since it seems that although authors perform a
calculation of the energy up to second order in the gas
parameter $x$ they only compute the free energy to first order in $x$, i.e. the
Hartree-Fock (which they use to determine the nature of the transition). This
would be an uncontrolled way of performing
perturbation theory and therefore puts under question the validity of the
findings. One needs to go to the appendix to realize that the different terms
are of order $O(x^2)$. This is however not clear from
the manuscript and needs to be clarified early on in section III.

Author: The Referee is spot-on. In the new version of the manuscript, Sec.
III, we
have made explicit and more clear that we first analyze the first order model,
where the coefficients do not depend on $x$, and then we report the
second order model, where now the coefficients depend on $x$.

Concern 3: In the introduction the authors should briefly discuss the
progress with ultracold atoms and SU(N) physics with AEAs in the last years.
Here I present some suggestions:\\
a) PRResearch 2, 012028(R)(2020) studied the collective excitations in SU(N)
Fermi gases.\\
b) PRL 121, 167205 (2018) studied interaction effects in SU(N) Fermi gases as N
is varied.\\
c) PRA 104, 043316 (2021) found universal behavior of thermodynamic quantities
with N in SU(N) gases in optical lattices.\\
d) PRA 101, 053620 (2020) studied prethermalization in SU(N) gases where
different initial conditions are studied before a quench.\\
e) Nat. Phys. 10, 779 (2014) was one of the first studies to demonstrate the
SU(N) symmetric interactions in AEAs.\\
f) arxiv:2010.07730 observed antiferromagnetic correlations in SU(N) gases in
optical lattices and achieved extremely low temperatures.\\
g) Nat. Phys. 16, 1216 (2020) perform a thorough study of the thermodynamics of
deeply degenerate SU(N) Fermi gases.

Author: We acknowledge the Referee for pointing us to these references. They
have been introduced in the new version. Also, we introduced references to
previous work by Cazalilla e al.

Concern 4: From Fig. 4 is unclear that the Stoner model predicts a
diverging $\chi$. I suggest making this explicit from eq.(22).

Author: This is now done explicitly.

Concern 5: Figure 5 is present in the manuscript, but is never discussed in
the manuscript. Either is not relevant and it should be deleted or proper
discussion should be presented.

Author: We made a mistake with the references. The reference to Fig 7, in the
end
of page 4, was to be to Fig 5. We have fixed it.

Concern 6: On page 5, first column authors mention:'In both cases, the
polarization jumps from $P=0$ to $1$ without intermediate values, in contrast to
the case of $s=1/2$'. This sentence is slightly unclear. Authors should mention
that is the minimum of the energy as a function of the polarization what jumps.
For example:'In both cases, the location of the minimum of the energy as a
function of polarization jumps from $P=0$ to $1$ without intermediate values, in
contrast to the case of $s=1/2$'.

Author: We have changed this part.

Concern 7: On page 6, authors mention 'In Fig. 9, we also plot experimental
results from Ref. [27] that show a similar behavior to the one observed with our
second order expansion (once properly rescaled)'.
Authors should give a brief explanation of how this rescaling was done.

Author: Observing the experimental values of Fig. 9 that come from Ref. 27, we
have
found a scaling factor of $(2\pi)^2$. This value comes from comparing their
definitions with ours being its origin a different normalization. %, however,
the rest of the normalization that they have done is quite obscure and we have
not been able to find the whole scaling factor. The origin of the problem is
that Ref. 27 does not use the standard definition and introduced additional
normalization factors. Some time ago, we contacted with their authors but we
were not able to arrive to a numerical agreement. Due to that, we have desisted
from comparing the values quantitatively and we only compare the qualitative
trend. The scaling factor that we apply is just a value to match their values
with ours. Approximately, the factor we apply is $\approx 0.2 (2\pi)^3$.

In the new version of the manuscript, we have added a brief comment about the
scaling and given the value of the scaling factor.

Concern 8: For large-spin, some interesting physics arises (such as
long-lived prethermal states) from initial conditions where there is a spin
imbalance in for the different spin projections [PRA 101, 053620 (2020)], as
opposed to the particular polarization case considered in the manuscript. Can
authors comment on how they expect their results to change when other
polarizations are considered?

Author: In our answer to Concern \#5 of the first Referee we have addressed
also
this comment (copied below). We must say that our formalism is completely
general and can work with any initial population distribution and polarization
protocol. However, these other cases are always of higher energy because we
have
proved that our work deals with the ground-state situation. Following the
advice of the Referee, we have added a comment in the first paragraph of Sec. V
about what
happens when other polarizations are considered.

We have examined the configuration used in Ref. [25]. They have a
system with $s=3/2$ and they set $N_{3/2}=N_{-3/2}$, $N_{1/2}=N_{-1/2}$ and
$N_{\pm1/2}=rN_{\pm3/2}$ where $r$ is 1.83. This initial configuration also has
zero polarization, but has a higher energy than ours. We have tried to polarize
this system using two polarization protocols.

In the first, we break the equality between $N_{3/2}$ and $N_{-3/2}$, and we
send $N_{-3/2}$, $N_{-1/2}$ and $N_{1/2}$ to zero as $N_{3/2}$ increases.
Although the initial configuration is different from ours, the system
transitions to $P=1$ around $x^*\approx0.94$. This value is very similar to the
one we have presented for $s=3/2$, $x_{our}^*\approx0.95$. It has transitioned
at a slightly lower value of $x^*$, but we recall that at $x^*0.94$ this case
has more energy. If we look at the Tan's contact (which we can do), the
behavior
is very similar to the one we have shown in Fig. 8. As long as the gas does not
transition, the Tan's contact increases. The quantitative values are obviously
different.

In the second protocol, we have kept the equalities between
$N_{3/2}$ and $N_{-3/2}$ and we have just increased the gas parameter. We have
seen that at $x=1.25$ the components $N_{\pm1/2}$ drop to zero abruptly and the
components $N_{\pm3/2}$ become $N/2$. After that, if we break the equality
between $N_{3/2}$ and $N_{-3/2}$ as we have done before because we want to
achieve $P=1$, the component $N_{-3/2}$ immediately drops to zero because at a
gas parameter of $x=1.25$ the gas wants to be fully-polarized. Concerning the
Tan's contact, it behaves as before and the case we have presented, as long as
it
does not transition, the Tan's contact increases, but with different
quantitative
values.

To sum up, if other polarization configurations are considered, the energy will
be higher because it will not be the ground state, the Tan's contact will also
increase as long as the system does not transition and the critical value $x^*$
at which the transition occurs will vary.

Concern 9: Reference [7] Valtolina et. al., Nat Phys 13 704 (2017) was
performed with $^6$Li atoms, which is the
fermionic isotope.

Author: We have amended that.

Concern 10: I strongly advice the use of visually perceptive colormaps for
Figures 3 and 6. The use of a visually perceptive colormap is not only useful if
the paper is printed in black and white, but carries informational content. For
example: In Figure 6 results of $E/N$ are presented as a function of $P$ for
different values of $x$ . Authors should use a visually perceptive colormap,
since this would immediately indicate the reader which curve corresponds to
smaller/larger value of $x$.

Author: We thank the Referee for the suggestion. We have done the changes in
the
figures.

Concern 11: In Figure 3, what is $c$? Should it be $x$ instead?

Author: Yes, our mistake. It is $x$. We have changed it.

Concern 12: On page4, last paragraph authors first mention Figure 7 and
then Figure 6. They should change the order of the figures to ensure a coherent
narrative.

Author: This reference to Fig 7 is the one we have pointed out before that it
was
meant to Fig 5. It was our misprint.

Concern 13: In Figures 3 and 6, authors should use a markers to indicate
the location of the minima for the different curves (helps with the discussion
on the main text).

Author: Done.

Concern 14: On Figure 10, have the authors tried fitting the data? Would be
interesting to know how does $x$ scale with N. In addition, presenting the
results from a large-$N$ expansion. Its validity can help diagnose where simple
approximations are still valid and can simplify their analytic expressions.

Author: We have found the analytical form of the transition points (see Eqs. 29
and 30). In Fig 10, we have added these equations as lines. These expressions
are only valid for $s>1/2$ because at the transition the polarization goes from
0 to 1 directly.

Concern 15: In the conclusions, authors mention 'Beyond the second-order
terms analyzed in this work, the energy ceases to be universal in terms of the
gas parameters because other scattering parameters of the interactions, mainly
the $s$-wave effective range and $p$-wave scattering length get involved.'
Given that ultracold atom experiments deal with low-energy physics, mainly
interactions are dominated by $s$-waves; authors should connect this to justify
why neglecting higher-order terms and/or higher partial-waves scattering lengths
is adequate.

Author: The introduction of second-order terms improves notably the predictions
about the ferromagnetic transition and other observables, mainly because at the
transition point the values of $x$ are relatively large. Beyond second-order,
other scattering parameters enter into play and one leaves the universality in
terms of solely the s-wave scattering length. As shown in Ref. 9, at the
transition point (s=1/2) the second-order approximation is not able to
reproduce accurately the QMC results. Using two different potentials (with same
scattering length) the QMC prediction on the transition density slightly
depends on the specific shape of the potentials. This means that other
parameters than the s-wave scattering length are necessary to account for the
results. Introduction of third-order (Ref. 14) improves the perturbative
results and reproduce accurately the QMC results.
In the new version of the manuscript, we have commented on that in the end of
the Conclusion section.

Concern 16: In the abstract:'This is an extension of an old result'
$\rightarrow$ 'This is an extension of previous results'.

Author: Done.

Concern 17: On page 2, second column:'We have been able to integrate Eq.(7)
and find an...' $\rightarrow$ 'We have been able to integrate Eq.(7) and found
an...'.

Author: Done.

Concern 18: On page 3, section IV:'formalism' $\rightarrow$ 'formalism'.

Author: Done.

Concern 19: On page 5, first column:'... we have access to other important
properties as the Tan's constant.' $\rightarrow$ '... we have access to other
important properties such as the Tan contact.'(is Tan contact rather
than Tan constant).

Author: Done.

Concern 20: On page 6, first column:'In other words, the Fermi gas becomes
more interactive and...' $\rightarrow$ 'In other words, the Fermi gas becomes
more interacting and...'

Author: Done.

Concern 21: Include a literature review in the introduction.

Author: In the new version we have enlarged the number of interesting
references,
following the advice of the Referee.

Concern 22: For Figs. 3, 6, 8 use a perceptive color map.

Author: Done.

Concern 23: For Figs. 3 and 6 also indicate with a marker the location of
the minima (for easier visualization purposes).

Author: Done.

---

## Round 2 · Referee Report · Anonymous (Referee 2) · 2022-7-1

Strengths

  • provides analytical insight into the ferromagnetism of SU(N) Fermi gases taking into account the polariation/spin imbalance

  • provides an improvement on prior work, in particular in regards to the transition point for s=1/2 mixtures

Weaknesses

  • the quantitative comparison could be more detailed, both in terms of experiment and other/prior theoretical approaches

  • the quantitative agreement or improvement over prior approaches is not immediately clear

Report

The manuscript presents analytical expressions for the energy of SU(N) Fermi gases to second order in the gas parameter, accounting for different spin channel occupations.

These results are used to discuss the transition from a paramagnetic to a ferromagnetic state as a function of the interaction strength, as well as to derive expressions for the spin susceptibility and tan's contact, which in principle can be probed in experimental settings.

The main novel predictions are a first order transition for s=1/2 (instead of the continuous one predicted by a first order calculation), as well as predictions for the behaviour of transition point and Tan's contact as a function of N.

Given the potential applicability to experimental work on SU(N) Fermi gases which is a rapidly growing field, I believe the work is suitable for publication in SciPost Phys after the following comments have been addressed.

Questions:

  • For the experimental comparison in Fig 9, what is the required rescaling factor? Depending on the magnitude of this, could the authors comment on what would explain the mismatch/whether they expect their results to be quantitatively correct, or rather mainly see the benefit in explaining the qualitative trends.

  • Related to above, could the authors comment on quantitative agreement/disagreement of their predictions, both with experimental values, as well as prior theoretical work for e.g. the phase transition point, or the values of the tan’s contact?

    • Further, if for s>1/2 the gas is always either fully polarised, e.g. non-interacting, or fully spin-balanced, what is the quantitative difference of the results of the current manuscript to a first order calculation, e.g. for the results in Fig 9 and 10.
  • Could the authors comment on the feasibility to test some of their predictions. For the contact which can be measured from my understanding their results below the transition reduce to the unpolarised case, and from what I understand therefore prior results (see question below). Is the transition as a function of x actually accessible in a cold gas experiment, where the polarisation/spin composition is essentially determined by the initial preparation, and cannot change/thermalise. I assume the authors might imagine experiments along the lines of Ref 7 to probe the spin susceptibility and the transition. In any case pointing out which and how their predictions can be tested might be useful to mention in the manuscript.

  • Related to this, could the authors comment on potentially other interesting cases where the polarisation is not just given by a majority spin and balanced minority spins. Since an initially prepared imbalance is (at least) meta-stable, presumably one could predict and measure a tan contact for such a non groundstate mixture as well.

  • The notation in Eq (14) and language after should be improved to clarify that the used symbols are not constants, but rather functions of x, and are explicitly of order 2 as provided in the appendix

Minor comments:

  • The sentence preceeding Eq. 9 refers to "its Landau expression" without ever saying what "it" refers to. The sentence also seems to claim Eq (9) is first order in x, when presumably it is second order?

  • For $\epsilon_F$ and $k_F$ could the authors give the explicit expressions to clarify whether they are defined for a spin-polarised gas of $N$ particles, or of $N/\nu$ particles?

  • Fig 5 is not referenced in the manuscript text. I believe the reference to Fig 7 in the last paragraph of page 4 right column should refer to Fig. 5 instead

  • The authors might consider the use of more clearly distinguishable color schemes in Fig 3,6, and/or distinguish curves in plots (Fig 1-8) in addition via different line types (dashed,dotted, etc), and different markers (circles/squares) for experimental and theory data in Fig 9.

Fig 9: - For the rescaling is this just an overall constant factor applied to the data. If so what is the scaling factor required to obtain agreement?

- Are these results consistent with/exactly the naive (N-1) scaling? The authors seem to scale by  $\nu = N$ instead in Fig 9. Would it be a straight line when scaling by (N-1) ? Since the results are analytic, what is the actual scaling with $\nu$, or N respectively?

Requested changes

  • extend the discussion of the quantitative comparison of their results to experimental data (e.g. in Fig 9, see also report), and/or prior theory results.

  • extend the discussion of which/how their predictions could be tested

  • minor changes as discussed in report

  • validity: good
  • significance: high
  • originality: good
  • clarity: good
  • formatting: good
  • grammar: good

Author:  Jordi Boronat  on 2022-10-27  [id 2958]

(in reply to Report 2 on 2022-07-01)

Author: We kindly thank the Referee for his/her positive comments and for
finding our manuscript appropriate for SciPost Phys. In the following, we
address the comments and
suggestions pointed out by the Referee and the changes introduced in the
manuscript accordingly.

Concern #1: For the experimental comparison in Fig 9, what is the required
rescaling factor? Depending on the
magnitude of this, could the authors comment on what would explain the
mismatch/whether they
expect their results to be quantitatively correct, or rather mainly see the
benefit in explaining the
qualitative trends.

Author: The normalization in the experimental work (Ref. 27) and, even the
definition that they use, differ from the standard one used by us. %Some time
ago, we contacted one of the authors of Ref. 27 and his response was rather
confusing.
Looking at Ref. 27, we have found that the conversion
from their measures to our predictions needs to multiply by a factor
$(2\pi)^2$. This value comes from comparing their definitions with ours. %But
this is not enough, and we have not been able to understand the remaining
factor. Due to that, we have desisted from comparing the values quantitatively
and we only compare the qualitative trend. The scaling factor that we apply is
just a value to match their values with ours, $\approx 0.2 (2\pi)^3$.

Concern 2: Related to above, could the authors comment on quantitative
agreement/disagreement of their
predictions, both with experimental values, as well as prior theoretical work
for e.g. the phase
transition point, or the values of the tan’s contact?

Author:For spin $1/2$, there are both experimental and theoretical
works, and they match with our results (see for instance the quantum Monte
Carlo results of Ref. 9). However, for spin $>1/2$, there are no previous
theoretical works that predict, for example, the phase-transition point,
magnetic susceptibilities or the Tan's contact In fact, there have not been
theoretical studies about the ferromagnetic transition (itinerant
ferromagnetism) for spin $>1/2$. And concerning experimental values, to the
best
of our knowledge, there are not experimental values other than the Tan's
contact
for comparing our results. We believe that our theoretical work is a first
exploration that can guide experimentalists towards the search of itinerant
ferromagnetism in SU(N) fermions.

Concern 3: Further, if for $s>1/2$ the gas is always either fully
polarized, e.g. non-interacting, or fully spin-balanced, what is the
quantitative difference of the results of the current manuscript to a first
order
calculation, e.g. for the results in Fig 9 and 10.

Author: We are working in a perturbative expansion in terms of the gas
parameter
$x$, therefore the difference between first order (Hartree Fock) and second
order increases with $x$. As the ferromagnetic transition happens at a
relatively large $x$, the correction of second-order is significant,
approaching for $s=1/2$ (the only studied case so far) quantum Monte Carlo
results. Following the advice of the Referee, we have added the Stoner
prediction to the results in Figs 9 and 10. Moreover,
we have included the lines with the equations that predict those values.

Concern 4: Could the authors comment on the feasibility to test some of
their predictions. For the contact which can be measured from my understanding
their results below the transition reduce to the
unpolarized case, and from what I understand therefore prior results (see
question below). Is the transition as a function of x actually accessible in a
cold gas experiment, where the polarization/spin
composition is essentially determined by the initial preparation, and cannot
change/thermalise. I assume the authors might imagine experiments along the
lines of Ref 7 to probe the spin susceptibility and the transition. In any case
pointing out which and how their predictions can be
tested might be useful to mention in the manuscript.

Author: The Referee is right on the Tan's contact: it is only measured
below the phase transition, because, after the transition, if the gas
fully-polarizes, it drops to zero.

According to experimental facts, there are two main problems to observe
itinerant ferromagnetism. The first one is that, upon increasing the
density of the system, the metastable repulsive gas decays to the lowest
energy states, forming molecules. The second problem is the one the Referee
points out. The transition is going to happen if the system can
thermalize and thus the different spin components can change, leading to the
ferromagnetic point where one spin component is maximized. Due to that, we
can imagine an experimental protocol similar to the one for spin 1/2 in Ref. 7,
but generalized to larger spins.

We have added a comment about how our predictions could be tested at the end of
the Results section.

Concern 5: Related to this, could the authors comment on potentially other
interesting cases where the polarization is not just given by a majority spin
and balanced minority spins. Since an initially prepared imbalance is (at least)
meta-stable, presumably one could predict and measure a tan contact for such a
non ground-state mixture as well.

Author: We thank the Referee for this interesting point. Remarkably, our
equations can be applied to any initial distribution of spin occupations and
to any protocol for reaching the ferromagnetic state. However, there are so
many options that we decided to focus only to the one corresponding to the
lowest energy configurations (ground state). Nevertheless, we have examined the
configuration used in Ref. [25]. They have a system with $s=3/2$ and they set
$N_{3/2}=N_{-3/2}$, $N_{1/2}=N_{-1/2}$, and $N_{\pm1/2}=rN_{\pm3/2}$, with
$r= 1.83$. This initial configuration has also zero polarization, but
has a higher energy than ours. We have polarized this system using two
protocols.

In the first one, we break the equality between $N_{3/2}$ and $N_{-3/2}$, and
we send $N_{-3/2}$, $N_{-1/2}$ and $N_{1/2}$ to zero as $N_{3/2}$ increases.
Although the initial configuration is different from ours, the system
transitions to $P=1$ around $x^*\approx0.94$. This value is very similar to the
one we have presented for $s=3/2$, $x_{our}^*\approx0.95$. It has transitioned
at a slightly lower value of $x^*$, but we recall that at $x^*0.94$ this case
has a larger energy. If we look at the Tan's contact (which we can do), the
behavior is very similar to the one that we have shown in Fig. 8. As long as
the gas does not transition, the Tan's contact increases. The quantitative
values
are obviously different.

And, in the second protocol, we have kept the equalities between
$N_{3/2}$ and $N_{-3/2}$ and we have just increased the gas parameter. We have
seen that at $x=1.25$ the components $N_{\pm1/2}$ drop to zero abruptly and the
components $N_{\pm3/2}$ become $N/2$. After that, if we break the equality
between $N_{3/2}$ and $N_{-3/2}$ as we have done before because we want to
achieve $P=1$, the component $N_{-3/2}$ immediately drops to zero because at a
gas parameter of $x=1.25$ the gas wants to be fully-polarized. Concerning the
Tan contact, it behaves as before and the case we have presented, as long as it
does not transition, the Tan's contact increases, but with different
quantitative
values.

An account of this consideration has been included in the first paragraph of the
Conclusions section.

Concern 6: The notation in Eq (14) and language after should be improved to
clarify that the used symbols are
not constants, but rather functions of x, and are explicitly of order 2 as
provided in the appendix.

Author: We have added the dependence with $x$ of the constants of the model to
distinguish Stoner and second order.

Concern 7: The sentence preceeding Eq. 9 refers to "its Landau expression"
without ever saying what "it"
refers to. The sentence also seems to claim Eq (9) is first order in x, when
presumably it is second order?

Author: We have rewritten 'the Landau expression of the energy'. Concerning the
second point, Eq. (14) (probably the Referee meant Eq. 14) is first order, while
Eq. (15) is second order.

Concern 8: For $\epsilon_F$ and $k_F$ could the authors give the explicit
expressions to clarify whether they are defined
for a spin-polarized gas of $N$ particles, or of $N/\nu$ particles?

Author: We have added the explicit expressions in the new version of the
manuscript (text before Eq. 1).

Concern 9: Fig 5 is not referenced in the manuscript text. I believe the
reference to Fig 7 in the last paragraph
of page 4 right column should refer to Fig. 5 instead.

Author: Thanks to the Referee for pointing to this error. We made a
mistake with the references. The reference to Fig 7 that the Referee mentions
was to be to Fig 5. We have fixed it.

Concern 10: The authors might consider the use of more clearly
distinguishable color schemes in Fig 3,6, and/or distinguish curves in plots
(Fig 1-8)in addition via different line types (dashed,dotted, etc), and
different markers (circles/squares) for experimental and theory data in Fig 9.

Author: We have changed the figures following the suggestion of the Referee.

Concern 11: {In Fig. (9): For the rescaling is this just an overall constant
factor applied to the data. If so what is the scaling factor required to obtain
agreement?

Author: Yes, it is just a constant factor, and it is $(2\pi)^2$. See also our
response to Concern 1

Concern 12: In Fig. (9): Are these results consistent with/exactly the
naive (N-1) scaling? The authors seem to scale by $\nu=N$ instead in Fig 9.
Would it be a straight line when scaling by (N-1) ? Since the results are
analytic, what is the actual scaling with $\nu$, or N respectively?

Author: First of all, we have changed slightly the definition of the Tan's
contact (Eq. 25) in order to be more consistent with previous papers. The Tan's
contact times the degeneracy goes as $(\nu-1)$, if we plot $C$, the dependence
now
will be $(1-1/\nu)$, that is why the points in Fig 9 get flattened when we
increase the degeneracy, so yes, they are consistent with the $\nu-1$ scaling.
The Referee is right when he/she says that it would be a straight line if the
scaling was made by $\nu-1$. However, we have decided to re-scale by $\nu$
because re-scaling by $\nu-1$ has problems at $\nu=1$, and also because the
experimental results were presented with this same scaling and we wanted to
keep the same shape.

Concern 13: Extend the discussion of the quantitative comparison of their
results to experimental data (e.g. in Fig 9, see also report), and/or prior
theory results.

Author: See our previous comments (Concern 4)

Concern 14: Extend the discussion of which/how their predictions could be
tested.

Author: As we have said before, we have added a explanation at the end of Sec.
V on how our predictions could be tested.

Concern 15: Minor changes as discussed in report.

Author: Done.

---

## Round 3 · Referee Report · Anonymous (Referee 2) · 2022-10-28

Report

In the revised version the authors address the issues raised in the first round of review satisfactorily, and I now recommend publication in SciPost Phys.

Requested changes

Clarification: - based on the author response to Concern #1 of report II it seems the data in Fig. 9 is rescaled by 0.2(2π)^2, but the manuscript still states "(2π)^2", e.g. in the caption of Fig.9 and in the paragraph just above the figure.

This should be clarified for the final version.

  • validity: -
  • significance: -
  • originality: -
  • clarity: -
  • formatting: -
  • grammar: -

Author:  Jordi Boronat  on 2022-11-03  [id 2977]

(in reply to Report 1 on 2022-10-28)

We kindly thank the Referee for for finding our manuscript appropriate for SciPost Phys. In the following, we address the comment pointed out by the Referee.

Concern 1:

Clarification: - based on the author response to Concern 1 of report II it seems the data in Fig. 9 is rescaled by $0.2(2\pi)^2$, but the manuscript still states "$(2\pi)^2$", e.g. in the caption of Fig. 9 and in the paragraph just above the figure. This should be clarified for the final version.

Author: We are sorry because we made a type editing mistake when we wrote the answers to the Referee report. The correct value that we use to scale the experimental points is $(2\pi)^2$, which is the value reported in the manuscript.

---

## Round 3 · Referee Report · Anonymous (Referee 1) · 2022-10-28

Strengths

1- The manuscript is very thorough and presents detailed calculations of the derived analytic expressions for the energy of repulsive Fermi gases up to 2nd order of perturbation theory in the gas parameter.

2- Provides clear insight into the nature of the ferromagnetic transition for $s\geq 1/2$.

Report

Authors have satisfactorily answered the questions in my report and incorporated the requested changes. In particular, the more detailed comparison against experiments and prior theoretical approaches, as well as the discussion with respect to previous theoretical results studying SU(N) imbalanced mixtures has strengthened the manuscript.

I recommend publication in SciPost Phys after the following minor comment is addressed:

  1. In Fig 10, why does the s=1/2 red cross do not match with the green line? I expect they should.

Requested changes

1- Explain/correct the discrepancy in Fig. 10.

  • validity: high
  • significance: high
  • originality: good
  • clarity: good
  • formatting: good
  • grammar: good

Author:  Jordi Boronat  on 2022-11-03  [id 2978]

(in reply to Report 2 on 2022-10-28)

We kindly thank the Referee for his/her positive comments and for finding our manuscript appropriate for SciPost Phys. In the following, we address the comment pointed out by the Referee and the changes introduced in the manuscript accordingly.

Concern 1
In Fig 10, why does the $s=1/2$ red cross do not match with the green line? I expect they should. Explain/correct the discrepancy in Fig. 10.

Author: The green and yellow lines in Fig. 10 are only valid when, at the transition, the polarization goes directly from 0 to 1; this is the case for $s>1/2$. However, at $s=1/2$ the Stoner model predicts a continuous transition, and the second-order approximation predicts a transition to a polarization of 0.545, hence the green and yellow lines do not apply at $s=1/2$. That is the origin of the discrepancy. This exception was already commented in the previous version ,below Eq. (30). However, and to be even more clear on this point, we have written the following additional sentence.

" Notice that for $s=1/2$ Eqs. (29) and (30) do not hold. "

---

## Round 3 · Author Response

Dear Editor:
Here we resubmit our manuscript with the changes suggested by the Referees.

J. Pera, J. Casulleras, J. Boronat

---

## Round 3 · List of Changes

This is reported in our replies to the Referees.

---

## Editorial Decision

published